# DNA methylation patterns facilitate tracing the origin of neuroendocrine neoplasms

Benjamin Goeppert [1,2,3] ✉, Alphonse Charbel [1,4], Reka Toth [5], Yue Zhang[5], Danial Tabbakh[5], Thomas Albrecht[1,4], Daniel Schrimpf[6,7,8], Louis de Mestier[9], Jérôme Cros [10], Monika Nadja Vogel[11,12], De-Hua Chang[4,13,14], Eva-Marie Bohn[1], Alexander Brobeil[1,15], Junfang Ji [16], Stephan Singer[17], Petr V. Nazarov [5], Aurel Perren [2], Leonidas Apostolidis [4,18], Andreas von Deimling [6,7,8] & Stephanie Roessler [1,4] ✉

Neuroendocrine neoplasms (NEN) are thought to originate from diffuse neuroendocrine networks and therefore most frequently arise in the gastrointestinal tract and lungs. The liver is a frequent site of metastasis of NEN but also the existence of primary hepatic NEN has been proposed. Due to the impact on disease management, it is urgently required to discriminate the origin of hepatic NEN metastases and to identify clinically relevant subgroups. Using a comprehensive set of NEN ($N = 212$) from two independent cohorts, we show that the DNA methylation profiles of NEN of distinct anatomical localizations differ significantly and primary tumor-metastasis pairs cluster together, enabling the identification of the tumor origin. Furthermore, the subgroup of hepatic NEN without clinically detectable primary tumor, thus classified as primary hepatic NEN, does not form a distinct cluster by DNA methylation analysis but colocalizes with various subgroups of extrahepatic NEN. Organ-specific subtyping of NEN delineates a foregut-like epigenetic profile for hepatic NEN with unknown primary. We propose a classifier with high prediction accuracy for each of the different organ sites. In conclusion, our results demonstrate that DNA methylation profiling enables precise prediction of NEN origin and suggests that a substantial proportion of presumed primary hepatic NEN may in fact represent misclassified secondary hepatic NEN of unknown primary.

Neuroendocrine neoplasms (NEN) may arise from endocrine glands or from diffuse neuroendocrine network systems of extraglandular tissue sites[1,2]. A rising incidence of NEN has been observed which increased 6.4-fold from 1973 to 2012 across all sites, stages and tumor grades, possibly due to improved diagnostic modalities and detection of early-stage disease[3–5]. NEN recapitulate various neuroendocrine cell types and are frequently categorized based on their location of origin, the foregut (including bronchopulmonal, stomach, duodenum and pancreas), the midgut (small bowel, appendix and proximal colon) or the

hindgut (distal colon and rectum), however, classification by organ, anatomical site and cell-of-origin could be clinically more relevant[3,6]. Apart from NEN developing in the lung, colorectum, appendix, duodenum and pancreas, NEN of other sites, e.g. skin or hepatobiliary system also occur but are exceedingly less frequent (Fig. 1A)[7–9]. The clinical presentation of NEN ranges from an indolent to a highly aggressive course with distant metastases and most patients presenting at late stages[10]. All high-grade NEN have a significantly higher risk of recurrence and dismal prognosis[5,11]. To date, the only curative

**Fig. 1 | Study design, sample and patient characterization. A** Schematic overview of the frequency of NEN developing in the lung, liver, pancreas, duodenum, appendix, colorectum and skin. This overview was created in BioRender. Roessler, S. (2025) https://BioRender.com/yn6lb0a. **B** A total of 212 NEN, consisting of the discovery set (Heidelberg cohort, $N = 197$ tissues from 185 patients) and the independent validation set (Beaujon cohort, $N = 15$ tissues from 14 patients) were included in this study. All NEN included in the Beaujon cohort were hepatic NEN without a known primary. The NEN of the Heidelberg cohort detected in the liver included hepatic NEN without known primary tumor ($N = 22$) and hepatic metastases of known primary NEN ($N = 22$), of which 9 were paired samples with the respective non-hepatic primary. A total of 65 gastrointestinal NEN, which included gastric/duodenal NEN ($N = 14$), ileal NEN ($N = 18$), appendiceal NEN ($N = 15$) and colorectal NEN ($N = 18$), were analyzed. In addition, 49 NEN of the lung, including pulmonary NEC ($N = 24$) and pulmonary carcinoids ($N = 25$), pancreatic NEN ($N = 25$) and Merkel cell carcinomas ($N = 14$ tissues of 11 patients) were included. **C–E** Immunohistochemical analysis of three representative hepatic NEN cases without known primary tumor (total of $N = 22$). Shown are H&E, chromogranin A (CHGA), synaptophysin (SYP) and Ki67 immunohistochemical stainings of one case with low (**C**), intermediate (**D**) and high (**E**) proliferation each. In total, 3 cases exhibited low, 5 cases intermediate and 14 cases high proliferation based on Ki67 staining (Table 1).

treatment option for NEN is surgery, but even after R0 resection, the recurrence rate is more than 40% and systemic treatment options for advanced NEN are limited[12]. Patients with gastrointestinal NEN may present with pain, bleeding, altered bowel habits, weight loss, anorexia or bowel obstruction and carcinoid syndrome may occur in patients with liver metastasis[1,13]. NEN are often discovered by detection of liver metastases, either because of related symptoms or incidentally detected during routine imaging examinations. In these cases, the primary tumor localization is most often detected during the initial

staging procedures, including radiological or nuclear imaging, such as computed tomography (CT), magnetic resonance imaging (MRI) or radioisotopic imaging including somatostatin receptor imaging (SRI) and 18fluoro-deoxyglucose (FDG) positron-emission tomography (PET). Clinical staging may be supplemented with digestive and/or bronchopulmonary endoscopy, conducted either as unsupervised exploratory procedures or directed by organ-specific immunostainings identified in the diagnostic liver biopsy. However, in 11–22% of patients, no primary tumor is found despite exhaustive clinical

investigation, previously resulting in the diagnosis of 'liver metastatic NEN of unknown primary'[4,14]. In the current WHO-classification, 5th Ed., the diagnostic entity of a primary hepatic neuroendocrine neoplasm was introduced for these cases[15]. In contrast, the previous WHO classification, 4th Ed., did not allow such a classification, as only metastases of NEN had been described and a primary hepatic neuroendocrine neoplasm could not be classified at that time. Currently, hepatic NEN are defined as hepatic epithelial neoplasms with morphological and immunohistochemical features of neuroendocrine differentiation including well-differentiated neuroendocrine tumors (NET) and poorly differentiated neuroendocrine carcinomas (NEC). Mixed-neuroendocrine and non-neuroendocrine neoplasms (MiNEN) have an NEN component and a non-neuroendocrine component, of which both have to account for ≥ 30% of the tumor. In hepatic MiNEN the non-neuroendocrine component may constitute hepatocellular carcinoma (HCC) or cholangiocarcinoma (CCA)[15]. However, apart from the clinical setting, well-established diagnostic guidelines for classifying hepatic neuroendocrine neoplasms in the differential diagnosis of metastasis of an extrahepatic NEN do not exist[16]. Due to the high histomorphological and genetic heterogeneity of ubiquitous NEN, it is urgently required to identify the tumor origin of hepatic NEN to enable correct diagnosis and appropriate therapy as response rates vary according to the origin of the primary tumor.

Identifying the primary tumor is highly clinically relevant in metastatic NEN as resection of the primary has prognostic impact. The therapeutic approaches for NEN metastasized to the liver include resection of all hepatic metastases, which may be curative[17]. Systemic therapeutic options for NET are somatostatin analogs, peptide receptor radionuclide therapy (PRRT), or everolimus; however, in pancreatic NET, other protocols such as sunitinib or streptozotocin-based chemotherapy have shown efficacy[17,18]. Regarding the aggressive nature of NEC and the common application of platinum- and etoposide-based first-line chemotherapy, one might assume primary identification to be less relevant[17]. However, immunotherapy should be added to chemotherapy for pulmonary NEC[19] and immunotherapy alone is preferred over chemotherapy in Merkel cell carcinoma[20]. Second-line treatment regimens differ greatly between NEC of different primary sites[19,21]. Management of primary hepatic NEN is performed in analogy to NEN with gastrointestinal primaries. Taken together, the detailed subtyping and identification of the primary is crucial for NEN treatment.

DNA methylation signatures have gained attention due to their capability of tracing cells-of-origin. This resulted in the recent development of promising algorithms to characterize cancers of unknown primary, brain tumors and sarcomas according to their epigenetic profiles[22–24]. Thus, we aimed to identify the molecular, in particular epigenetic alterations, involved in NEN development and specific to NEN subgroups. As systematic epigenetic studies on these rare hepatic NEN have not yet been performed, we here characterized the DNA methylation pattern of hepatic NEN and compared these to the liver metastases of NEN from the most common localizations. In addition, we compared our results to a comprehensive cohort of well-defined NEN primaries of various sites, localized to the pancreas, ileum, appendix, colorectum and lung. Furthermore, deconvolution analysis identified latent methylation components (LMC) and potential tumor origin sites, thereby providing a classification by DNA methylation profiling. In particular, DNA methylation analysis could identify the extrahepatic origin of a substantial proportion of hepatic NEN that were clinicopathologically classified as primary hepatic NEN.

## Results
### Clinicopathological characteristics of the study cohort
This study included a total of 212 NEN tissues, the discovery set consisted of 197 tissues from 185 patients (Heidelberg cohort) and the independent validation cohort (Beaujon cohort) included 15 well-

**Table 1 | Characteristics of patients with hepatic NEN or NEN liver metastases (N = 44)**

| Parameter | | Hepatic NEN | NEN liver metastases | p-value* |
|---|---|---|---|---|
| **Total** | **Number (percent)** | 22 (100.0) | 22 (100.0) | |
| **Age [years]** | Median | 64.5 | 54.5 | **0.013** |
| | Mean | 64.6 | 51.3 | |
| | Range | 35–83 | 14-82 | |
| | 95% Confidence interval (CI) | 59.0–70.1 | 42.9–59.6 | |
| **Sex** | male | 16 (72.7) | 9 (40.9) | 0.067 |
| | female | 6 (27.3) | 13 (59.1) | |
| **Primary** | Pancreas | 0 (0.0) | 7 (31.8) | **<0.001$** |
| | Ileum | 0 (0.0) | 8 (36.4) | |
| | Colon | 0 (0.0) | 1 (4.5) | |
| | Duodenum | 0 (0.0) | 2 (9.1) | |
| | Pulmonary NEC, SCNEC+ | 0 (0.0) | 3 (13.6) | |
| | Atypical pulmonary carcinoid | 0 (0.0) | 1 (4.5) | |
| | NA# | 22 (100.0) | 0 (0.0) | |
| **Surgical procedure** | Resection/transplantation | 7 (31.8) | 14 (63.6) | 0.069 |
| | Biopsy | 15 (68.2) | 8 (36.4) | |
| **Systemic and locoregional therapy** | Targeted** | 3 (13.6) | 5 (22.7) | 0.275 |
| | Chemotherapy | 10 (45.5) | 12 (54.5) | |
| | Somatostatin analogs | 3 (13.6) | 11 (50.0) | |
| | PRRT*** | 1 (4.5) | 6 (27.3) | |
| | SIRT**** | 0 (0.0) | 3 (13.6) | |
| **G** | NET G1 | 3 (13.6) | 5 (22.7) | 0.078 |
| | NET G2 | 5 (22.7) | 9 (40.9) | |
| | NET G3 | 3 (13.6) | 4 (18.2) | |
| | SCNEC+ | 2 (9.1) | 3 (13.6) | |
| | LCNEC+ | 9 (40.9) | 1 (4.5) | |
| **Ki67** | <3% | 3 (13.6) | 5 (22.7) | 0.261 |
| | 3–20% | 5 (22.7) | 9 (40.9) | |
| | >20% | 14 (63.6) | 8 (36.4) | |

*The two-sided Mann-Whitney U test, Fisher's exact or Chi-square test was performed as appropriate; ** Some patients received multiple therapies. Targeted therapy included: Everolimus, antiangiogenetic drugs or immune checkpoint inhibitors; ***Peptide receptor radionuclide therapy (PRRT); ****Selective internal radiation therapy (SIRT); + Small-cell neuroendocrine carcinoma: SCNEC. Large-cell neuroendocrine carcinoma: LCNEC; $ The exact value cannot be shown as it is less than <1E–10; # NA: not available; Bold values indicate statistical significance p < 0.05.

characterized hepatic NEN of 14 patients treated and diagnosed at Beaujon University Hospital, Paris, France[25]. NEN of the discovery cohort localized to the liver, pancreas, stomach, duodenum, ileum, appendix, colorectum, lung, and skin (Fig. 1B, Table 1, Table 2). In detail, the NEN samples included hepatic NEN without known non-hepatic primary tumor (N = 22) and liver metastases of known primary NEN (N = 22) of which 9 were paired samples with the respective non-hepatic primary NEN. In addition, gastric/duodenal NEN (N = 14), ileal NEN (N = 18), appendiceal NEN (N = 15), colorectal NEN (N = 18), pancreatic NEN (N = 25), pulmonary NEC (N = 24), pulmonary carcinoids (N = 25) and Merkel cell carcinoma (14 tissue samples of 11 patients) were included (Fig. 1B, Table 1, Table 2). Colorectal NEN included 9 NEN of the ascending colon, 1 NEN of the transverse colon, 3 NEN of the sigmoid colon and 5 NEN of the rectum. In summary, the study cohort (Heidelberg cohort) comprised a total of 197 NEN tissue samples of 185

**Table 2 | Patient characteristics of patients (N = 150) with NEN other than liver NEN, including 12 patients with MiNEN (see Table 3 for MiNEN)**

| Parameter | | Appendix | Ileum | Gastric/ duodenal | Colorectal | Pancreas | Pulmonary NEC | Pulmonary carcinoid | Merkel cell carcinoma | p-value* |
|---|---|---|---|---|---|---|---|---|---|---|
| **Total** | **Number (percent)** | **15 (100.0)** | **18 (100.0)** | **14 (100.0)** | **18 (100.0)** | **25 (100.0)** | **24 (100.0)** | **25 (100.0)** | **11 (100.0)** | |
| **Age [years]** | *Median* | 40.0 | 69.0 | 63.5 | 64.5 | 61.0 | 64.5 | 61.0 | 74.0 | **0.001** |
| | *Mean* | 46.5 | 64.9 | 65.6 | 63.8 | 57.5 | 63.4 | 58.0 | 70.9 | |
| | *Range* | 14–88 | 43–82 | 47–91 | 31–92 | 24–82 | 56–71 | 22–81 | 55–80 | |
| | *95% CI*** | 32.9–60.1 | 58.8–71.1 | 58.0–73.1 | 55.6-72.1 | 51.2–63.8 | 60.0–66.2 | 51.8–64.3 | 65.1–76.7 | |
| **Sex** | *male* | 7 (46.7) | 12 (66.7) | 7 (50) | 14 (77.8) | 16 (64.0) | 13 (54.2) | 11 (44.0) | 8 (72.7) | 0.322 |
| | *female* | 8 (53.3) | 6 (33.3) | 7 (50) | 4 (22.2) | 9 (36.0) | 11 (45.8) | 14 (56.0) | 3 (27.3) | |
| **pT$** | *pT1* | 11 (73.3) | 0 (0.0) | 3 (21.4) | 4 (22.2) | 3 (12.0) | 3 (12.5) | 16 (64.0) | 0 (0.0) | **<0.001^** |
| | *pT2* | 1 (6.7) | 3 (16.7) | 4 (28.6) | 1 (5.6) | 3 (12.0) | 3 (12.5) | 1 (4.0) | 0 (0.0) | |
| | *pT3* | 1 (6.7) | 9 (50.0) | 4 (28.6) | 6 (33.3) | 18 (72.0) | 1 (4.2) | 4 (16.0) | 0 (0.0) | |
| | *pT4* | 0 (0.0) | 5 (27.8) | 2 (14.3) | 6 (33.3) | 1 (4.0) | 5 (20.8) | 1 (4.0) | 0 (0.0) | |
| | *NA* | 2 (13.3) | 1 (5.6) | 1 (7.1) | 1 (5.6) | 0 (0.0) | 12 (50.0) | 3 (12.0) | 11 (100.0) | |
| **pN$** | *pN0* | 1 (6.7) | 1 (5.6) | 3 (21.4) | 2 (11.1) | 13 (52.0) | 6 (25.0) | 16 (64.0) | 0 (0.0) | **<0.001^** |
| | *pN1* | 0 (0.0) | 14 (77.8) | 6 (42.9) | 7 (38.9) | 9 (36.0) | 1 (4.2) | 0 (0.0) | 0 (0.0) | |
| | *pN2* | 0 (0.0) | 1 (5.6) | 1 (7.1) | 6 (33.3) | 0 (0.0) | 5 (20.8) | 4 (16.0) | 0 (0.0) | |
| | *pN3* | 0 (0.0) | 0 (0.0) | 1 (7.1) | 1 (5.6) | 0 (0.0) | 0 (0.0) | 0 (0.0) | 0 (0.0) | |
| | *NA* | 14 (93.3) | 2 (11.1) | 3 (21.4) | 2 (11.1) | 3 (12.0) | 12 (50.0) | 5 (20.0) | 11 (100.0) | |
| **M$** | *M0* | 1 (6.7) | 3 (16.7) | 0 (0.0) | 0 (0.0) | 2 (8.0) | 0 (0.0) | 0 (0.0) | 0 (0.0) | NA# |
| | *M1* | 0 (0.0) | 8 (44.4) | 2 (14.3) | 2 (11.1) | 5 (20.0) | 0 (0.0) | 1 (4.0) | 0 (0.0) | |
| | *NA* | 14 (93.3) | 7 (38.9) | 12 (85.7) | 16 (88.9) | 18 (72.0) | 24 (100.0) | 24 (96.0) | 11 (100.0) | |
| **G** | *NET G1* | 15 (100.0) | 9 (50.0) | 8 (57.1) | 4 (22.2) | 8 (32.0) | 0 (0.0) | 15 (60.0)§ | 0 (0.0) | **<0.001^** |
| | *NET G2* | 0 (0.0) | 8 (44.4) | 1 (7.1) | 2 (11.1) | 11 (44.0) | 0 (0.0) | 10 (40.0)§ | 0 (0.0) | |
| | *NET G3* | 0 (0.0) | 1 (5.6) | 1 (7.1) | 1 (5.6) | 6 (24.0) | 0 (0.0) | 0 (0.0) | 0 (0.0) | |
| | *SCNEC+* | 0 (0.0) | 0 (0.0) | 4 (28.6) | 2 (11.1) | 0 (0.0) | 14 (58.3) | 0 (0.0) | 11 (100.0)x | |
| | *LCNEC+* | 0 (0.0) | 0 (0.0) | 0 (0.0) | 9 (50.0) | 0 (0.0) | 10 (41.7) | 0 (0.0) | 0 (0.0) | |
| **Ki67** | <3% | 15 (100.0) | 9 (50.0) | 8 (57.1) | 4 (22.2) | 8 (32.0) | 0 (0.0) | 11 (44.0) | 0 (0.0) | **<0.001^** |
| | 3-20% | 0 (0.0) | 8 (44.4) | 1 (7.1) | 2 (11.1) | 11 (44.0) | 0 (0.0) | 14 (56.0) | 0 (0.0) | |
| | >20% | 0 (0.0) | 1 (5.6) | 5 (35.7) | 12 (66.7) | 6 (24.0) | 24 (100.0) | 0 (0.0) | 11 (100.0) | |

*Missing values and groups with more than 50% missing values were not included in the statistical analysis. The two-sided ANOVA or Chi-square test was performed as appropriate; **95% CI: 95% Confidence interval; ^The exact value cannot be shown as it is less than <1E−10; $ TNM classification of the primary NEN; for hepatic NEN and in some cases of metastatic NEN, TNM could only be performed in resection cases. # NA: not available; § Typical carcinoid of the lung (G1) and atypical carcinoid of the lung (G2); + Small-cell neuroendocrine carcinoma: SCNEC; large-cell neuroendocrine carcinoma: LCNEC; x Merkel cell carcinoma was graded as SCNEC but not included in the statistical analysis; Bold values in the table indicate statistical significance p < 0.05.

patients. Non-hepatic origin of hepatic NEN was ruled out in the clinical setting including CT and/or MRT imaging, in combination with endoscopy. Liver resection was performed for 7 (31.8%) and biopsy was taken for 15 (68.2%) of the 22 patients to confirm the initial diagnosis of liver cancer (Table 1).

All hepatic NEN were characterized by immunohistochemical staining against chromogranin A (CHGA) and synaptophysin (SYP) to confirm neuroendocrine differentiation (Fig. 1C–E). Detailed characterization of pulmonary NEC, pulmonary carcinoid, pancreatic NEN, ileal NEN, and MiNEN was performed using immunohistochemical staining to detect pan-cytokeratin (panCK), Thyreoglobulin (TG), Calcitonin (CT) and the transcription factors CDX2, SATB2, TTF1, GATA3, ARX and PDX1 (Supplementary Data 1). For immunohistochemical evaluation and molecular profiling of MiNEN, only the neuroendocrine component was analyzed. Furthermore, status of SSTR2A, RB1 and p53 were analyzed in NET G3 and NEC samples (Supplementary Data 2). Hepatic NEN of the discovery and independent validation cohorts were subjected to immunohistochemical analyses of these markers (Supplementary Data 3, Supplementary Data 4). Histopathological classification of NEN was based on immunohistochemical analyses of these extensive transcription factor analyses. In addition, tumor cell proliferation was assessed by anti-Ki67 immunohistochemical staining which varied from low

(< 3%) to intermediate (3-20%) and high (>20%) proliferation rate (Fig. 1C–E). Of the 22 hepatic NEN, 14 (63.6%) exhibited Ki67-positive staining of more than 20% of tumor cells, 5 (22.7%) cases had between 3 and 20% Ki67-positive tumor cells and the remaining 3 (13.6%) cases had less than 3% Ki67-positive tumor cells (Table 1). In addition, 9 (40.9%) of the hepatic NEN were large-cell neuroendocrine carcinomas (LCNEC), whereas the NEN liver metastases include only 1 (4.5%) LCNEC (Table 1). As expected, patient characteristics including age, pT, pN, tumor grading and proliferation rate (Ki67) differed significantly between NEN entities of different organs (Table 2). Two gastric/duodenal and 10 colorectal NEN were further classified as MiNEN (Table 3). The NEN component of MiNEN primarily exhibited high proliferation rates with more than 20% Ki67-positive tumor cells (91.7%, 11 out of 12 cases) and were classified as NET G3 (N = 2), small-cell carcinoma (SCNEC; N = 3) or large-cell carcinoma (LCNEC, N = 6), according to histomorphology and immunohistochemical analysis (Table 3). Overall survival was available for subsets of patients with hepatic NEN without detected primary tumor (N = 14) and liver metastasis of known NEN (N = 20). Hepatic NEN without detectable primary tumor had worse overall survival compared to patients with known NEN and liver metastasis (log-rank (Mantel-Cox) test, p = 0.005; Supplementary Fig. 1A). However, this can be accounted to the higher percentage of NEC in the hepatic NEN group, as hepatic

**Table 3 | Patient characteristics of patients with MiNEN (N = 12)**

| Parameter | | MiNEN |
|---|---|---|
| Total | Number (percent) | 12 (100.0) |
| Age | *Median* | 73.5 |
| | *Mean* | 71.3 |
| | *Range* | 53-92 |
| Sex | *male* | 10 (83.3) |
| | *female* | 2 (16.7) |
| Localization | *Gastric/duodenal* | 2 (16.7) |
| | *Colorectal* | 10 (83.3) |
| pT | *pT1* | 2 (16.7) |
| | *pT2* | 1 (8.3) |
| | *pT3* | 3 (25.0) |
| | *pT4* | 6 (50.0) |
| pN | *pN0* | 3 (25.0) |
| | *pN1* | 3 (25.0) |
| | *pN2* | 5 (41.7) |
| | *pN3* | 1 (8.3) |
| M | *M0* | 0 (0.0) |
| | *M1* | 2 (16.7) |
| | *NA* | 10 (83.3) |
| G | *NET G1* | 0 (0.0) |
| | *NET G2* | 1 (8.3) |
| | *NET G3* | 2 (16.7) |
| | *SCNEC* | 3 (25.0) |
| | *LCNEC* | 6 (50.0) |
| Ki67 | <3% | 0 (0.0) |
| | 3-20% | 1 (8.3) |
| | >20% | 11 (91.7) |

NEC showed an even stronger survival difference compared to hepatic NET (Table 1; Supplementary Fig. 1B).

## DNA methylation-based analysis shows distinct grouping for NEN of most organ sites

As DNA methylation patterns differ greatly between tumors of different tissues of origin, we focused our analysis on DNA methylation patterns of NEN of different organ sites, which show a large overlap with transcription factor expression. For visualization of high-dimensional data, t-SNE plots assign each sample a location in a two-dimensional map in such a way that similar samples are modeled by nearby points and dissimilar samples are modeled by distant points. For this purpose, we used our comprehensive NEN reference cohort ($N = 153$) including Merkel cell carcinoma ($N = 14$ of 11 patients), pulmonary NEC ($N = 24$), pulmonary carcinoids ($N = 25$), appendiceal NEN ($N = 15$), ileal NEN ($N = 18$), gastric/duodenal NEN ($N = 14$), colorectal NEN ($N = 18$) and pancreatic NEN ($N = 25$). In addition, we included HCC (TCGA-LIHC, $N = 50$)[26] and CCA ($N = 27$)[27] as a representation of the two most frequent primary liver cancers. Merkel cell carcinoma, pulmonary NEC, pulmonary carcinoids, appendiceal, ileal and pancreatic NEN clearly separated into distinct clusters of the resulting t-SNE plot (Fig. 2A). However, gastric/duodenal ($N = 14$) and colorectal NEN ($N = 18$) did not exhibit distinct clusters in the t-SNE plot and were therefore combined into the group of intestinal NEN not otherwise specified ($N = 32$, Fig. 2A). In detail, the 2 out of 9 NEN of the ascending colon clustered together with ileal NEN, whereas 1 NEN of the ascending colon showed appendiceal similarity and the remaining did not exhibit any clear clustering (Supplementary Fig. 2A). In line with these findings, the 12 MiNEN (2 gastric/duodenal and 10 colorectal MiNEN) did not fall into a distinct group (Supplementary Fig. 2B). Of

note, we only analyzed the neuroendocrine component of the MiNEN and for all cases the non-neuroendocrine tumor was adenocarcinoma of intestinal type and not further analyzed here. Similarly, restricting the analysis to the reference NEN only, excluding the hepatic NEN and NEN liver metastases, we observed that the Merkel cell NEN, pulmonary carcinoids, pulmonary NEC, appendiceal NEN, ileal NEN and pancreatic NEN fell into distinct groups but the gastric/duodenal and colorectal NEN and MiNEN respectively did not (Supplementary Fig. 2C, D). The colonic origin of these 9 NEN was confirmed by anti-SATB2 immunohistochemical staining (Supplementary Fig. 2E–G).

Hepatic NEN without known primary ($N = 22$) did not form a separate cluster but rather grouped within NEN clusters of other distinct organ sites, similar to hepatic metastases with known primary NEN ($N = 22$; Fig. 2A). To confirm that primary tumor and paired metastasis share similar epigenetic profiles, we focused on our paired sample set of primary NEN and corresponding metastases. Unsupervised hierarchical clustering and visualization by heatmap demonstrated high similarity in their DNA methylation profiles (Supplementary Fig. 3). To further elucidate the similarity between the paired samples, we visualized the 12 pairs in the t-SNE plot which revealed that 10 out of the 12 (83.3%) pairs clustered together, whereas the two samples of patient #8 and patient #9 fell into two different clusters in the two-dimensional t-SNE plot (Fig. 2B). While the primary pancreatic NEN of patient #9 clustered with other pancreatic NEN, the hepatic NEN resected one year later did not fall into this cluster (Fig. 2B, C). Comparative SNP analysis was conducted revealing high similarity for all paired samples which confirmed sample identity and indicated molecular changes associated with tumor progression (Supplementary Fig. 4). Similarly, the primary pancreatic NEN of patient #8 clustered with pancreatic NEN but the metastasis did not (Fig. 2B, C). Thus, most paired samples exhibited large similarities as 10 out of 12 pairs clustered in the t-SNE plot. Interestingly, hepatic NEN did not show a distinctive clustering as several hepatic NEN clustered together with various other extrahepatic NEN, such as pancreatic NEN or pulmonary NEC (Fig. 2C, D). As grading and Ki67-based proliferation rates significantly differ between NEN entities, we sought to investigate if the hepatic NEN also separated according to their rate of Ki67-positive cell counts (Table 2). Consistently, we observed that both extrahepatic and hepatic NEN with high proliferation rates, indicated by more than 20% Ki67-positive tumor cells, separated from NEN with low or intermediate proliferation rates (Ki67 < 20%, Fig. 2E, F). Moreover, NEC of small- and large-cell type clustered together, whereas NET G3 showed a heterogeneous profile (Supplementary Fig. 5). Thus, most NEN of different organ sites clearly separated into distinct clusters and proliferation status is associated with distinct epigenetic profiles of different NEN entities.

## Epigenetic profiling reveals distinct clusters of extrahepatic NEN, hepatic NEN, HCC and CCA

To decipher the origin of NEN within the liver, we studied the DNA methylation profiles of hepatic NEN with no primary tumor detectable elsewhere, therefore considered primary hepatic NEN ($N = 22$), hepatic metastases of known primary NEN ($N = 22$), paired HCC and non-neoplastic adjacent liver samples (TCGA-LIHC, $N = 50$)[26], CCA ($N = 27$)[27] and non-neoplastic bile duct controls ($N = 50$)[27,28]. The resulting t-SNE plot showed clear separation between HCC, CCA, non-neoplastic liver and non-neoplastic bile duct controls (Fig. 3A). Interestingly, hepatic NEN and NEN liver metastases split into two t-SNE clusters (Cluster 1 and Cluster 2; Fig. 3A), which did not overlap with HCC, CCA nor non-neoplastic controls. Each of the latter clusters contained both, cases of hepatic NEN and cases with NEN liver metastasis, demonstrating that hepatic NEN and NEN liver metastases have overlapping DNA methylation profiles (Fig. 3A).

As we have seen a separation based on proliferation status of the NEN of different organ sites (Fig. 2E, F), we analyzed the Ki67 status of

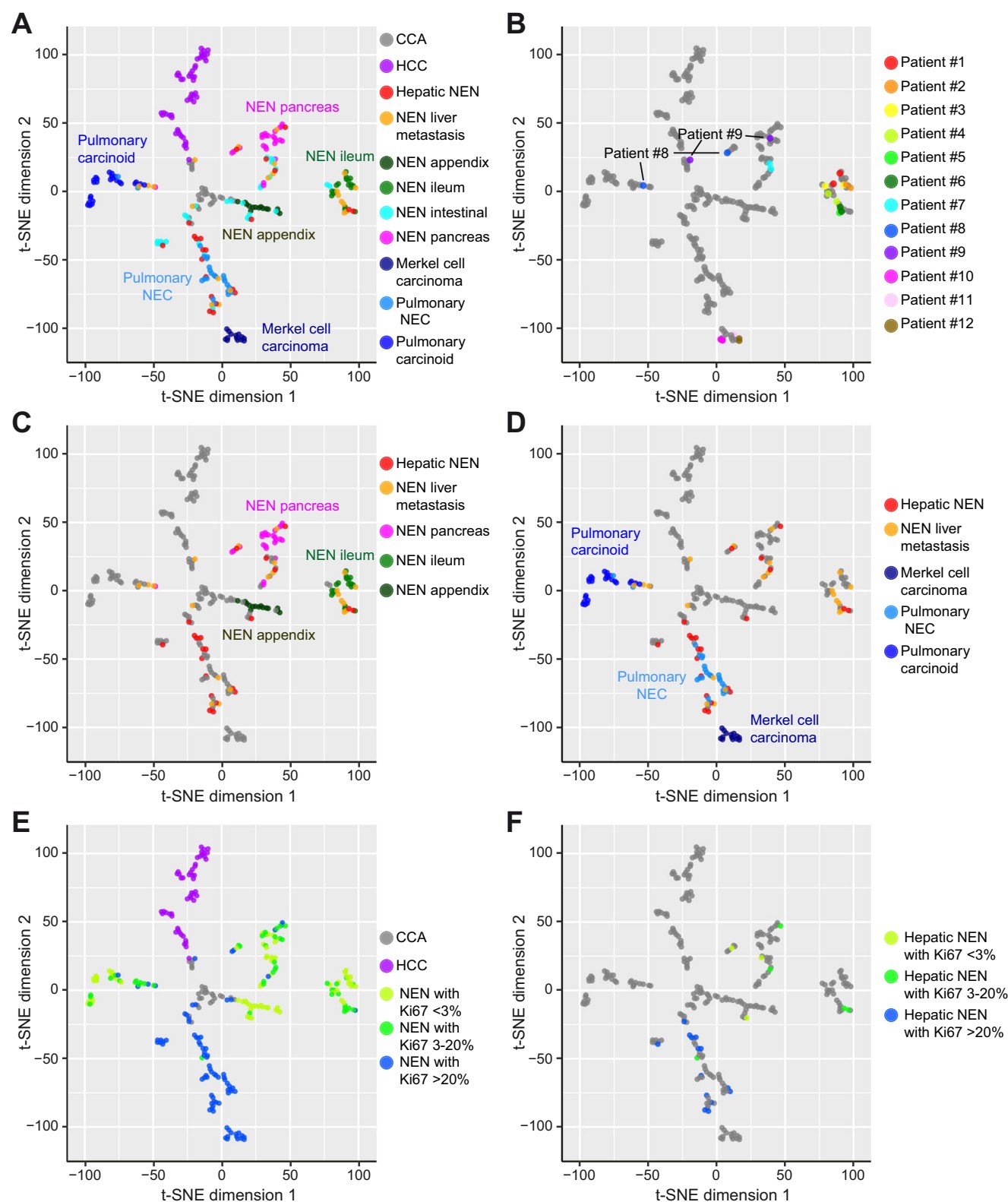

the hepatic NEN and NEN liver metastases further. We found that hepatic NEN and NEN liver metastases of Cluster 1 and Cluster 2 differed by their proliferation status (Fig. 3B). Well-differentiated NET G3 and poorly differentiated NEC, both with high proliferation rates of more than 20% Ki67-positive cells, separated from NET G1 (less than 3% Ki67-positive tumor cells) and NET G2 (3–20% Ki67-positive tumor cells; Fig. 3B). Cluster 1 included one single NET G2 with 15% Ki67-positive tumor cells, 7 NET G3 and 13 NEC, whereas Cluster 2 contained

8 NET G1, 13 NET G2 and 2 NET G3 (Chi-square test, $p < 0.001$). Thus, hepatic NEN and NEN liver metastases exhibit epigenetic profiles distinct from HCC and CCA and can be separated into a group with high proliferation and a second group with low to moderate proliferation.

Genomic instability is a hallmark of cancer implicated in cancer initiation, progression and therapy resistance. Patterns of copy number alterations (CNA) differ between cancer entities and are associated with specific oncogenes and tumor suppressors involved in the

**Fig. 2 | DNA methylation analysis reveals overlapping patterns of NEN liver metastases and hepatic NEN without known primary.** Shown are t-SNE plots of the DNA methylation profiles of CCA, HCC and a total of 197 NEN samples from different organ sites. **A** Indicated are all tumor subgroups including CCA ($N = 27$), HCC ($N = 50$), hepatic NEN without known primary tumor ($N = 22$), NEN liver metastases of known primary ($N = 22$), appendiceal NEN ($N = 15$), ileal NEN ($N = 18$), intestinal NEN not otherwise specified ($N = 32$), pancreatic NEN ($N = 25$), Merkel cell carcinomas ($N = 14$), pulmonary NEC ($N = 24$) and pulmonary carcinoids ($N = 25$). As only ileal but not gastric/duodenal ($N = 14$) and colorectal NEN ($N = 18$) exhibited a

clear cluster, gastric/duodenal and colorectal NEN were combined as intestinal NEN not otherwise specified. **B** Paired primary and metastasis samples of 12 patients are depicted. **C** Hepatic NEN showed overlapping DNA methylation patterns with pancreatic NEN and ileal NEN. **D** Furthermore, several hepatic NEN clustered together with pulmonary NEC. **E** Across NEN tumor entities, NEN with low (<3%) or intermediate (3–20%) percentage of Ki67-positive tumor cells separated from NEN with high Ki67 (>20%). **F** Hepatic NEN ($N = 22$) with low (<3%) or intermediate (3–20%) Ki67 expression clustered separately in comparison to those with high (>20%) Ki67 expression. CCA cholangiocarcinoma, HCC hepatocellular carcinoma.

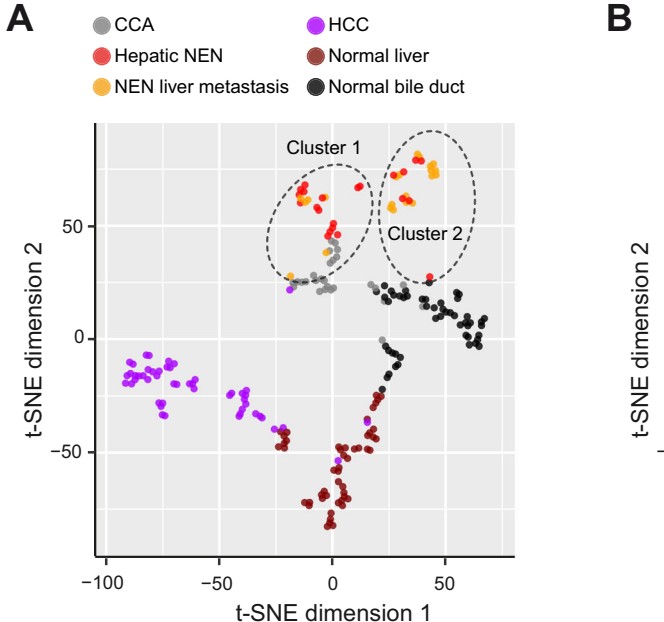

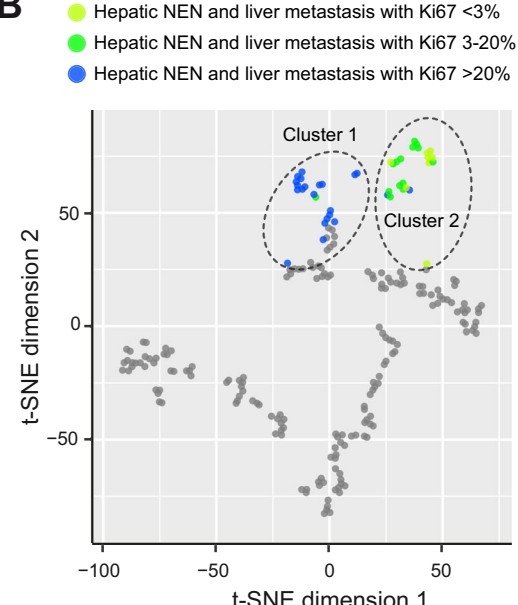

**Fig. 3 | DNA methylation analysis reveals distinct clusters of hepatic NEN, HCC and CCA. A** t-SNE plot of the DNA methylation profiles of CCA ($N = 27$), HCC ($N = 50$), hepatic NEN ($N = 22$), NEN liver metastases ($N = 22$), non-neoplastic (normal) liver ($N = 50$) and normal bile duct tissue samples ($N = 50$) showing a clear

separation of hepatic NEN from HCC, CCA and non-neoplastic samples ($N = 50$). **B** t-SNE plot with indication of hepatic NEN and NEN liver metastases with low (<3%), intermediate (3–20%) or high (>20%) percentage of Ki67-positive tumor cells. CCA cholangiocarcinoma, HCC hepatocellular carcinoma.

entities[29]. First, we compared NET G1, NET G2, NET G3 and NEC independent of the organ site. Consistent with previous reports, we found that NET G1 harbor only few genomic alterations, whereas NET G2 and NET G3 have progressively higher numbers of genomic alterations (Supplementary Fig. 6A–C)[30]. Furthermore, NET G3 and NEC differ in the affected chromosomes as chromosome 3q amplification only occurred in NEC underpinning the subtyping and histological characterization (Supplementary Fig. 6D–F, Supplementary Data 2). To dissect the genomic alterations in our NEN cohort, we performed copy number analysis of hepatic NEN without known primary tumor, liver metastases of known primary NEN, gastric/duodenal, ileal, appendiceal, colorectal and pancreatic NEN as well as pulmonary NEC, pulmonary carcinoids and Merkel cell carcinoma. Hepatic NEN and NEN liver metastases exhibited significantly different CNA profiles from CCA and HCC (Supplementary Fig. 7) with hepatic NEN exhibiting a large number of genomic alterations across the whole genome. Appendiceal NEN displayed minor to almost no CNA, whereas gastric/duodenal NEN, ileal NEN, colorectal NEN, pancreatic NEN, pulmonary NEC and pulmonary carcinoids exhibited varying degrees of alterations at distinct chromosomal locations (Fig. 4, Supplementary Fig. 8). Colorectal MiNEN displayed a tendency of a higher number of copy number losses compared to colorectal non-MiNEN (Supplementary Fig. 8). Focusing on pancreatic NEN and liver metastases of pancreatic NEN, we observed similar CNA profiles (Fig. 4A, B). Analogously, NEN of the ileum and liver metastases of ileal NEN or pulmonary NEC and

pulmonary NEC liver metastases displayed similar genomic alterations (Fig. 4C–F). Therefore, the genomic alterations of NEN of different organ sites exhibit organ-site specific CNA suggesting that CNA may provide a clue to the origin of NEN in addition to the epigenetic profiles.

## Most hepatic NEN are predicted to be of non-hepatic origin and show a foregut methylation pattern

We used deconvolution analysis to decompose methylation data into latent methylation components (LMC) and analyzed the proportions of the resulting LMC in each sample[31,32]. This resulted in 10 LMC with varying enrichment in Merkel cell carcinoma, appendiceal NEN, colorectal NEN, gastric/duodenal NEN, ileal NEN, hepatic NEN, NEN liver metastases, pancreatic NEN, pulmonary NEC and pulmonary carcinoids (Supplementary Fig. 9). Overall the 10 LMC exhibited little correlation with each other but LMC2 strongly correlated with Leukocytes Unmethylation for Purity (LUMP) which corresponds to tumor purity and was therefore excluded from further analyses (Fig. 5A, Supplementary Data 5). In addition, LMC2 did not exhibit a clear enrichment in any of the NEN subgroups (Supplementary Fig. 9). Most of the 9 remaining LMC showed specific association with a single NEN subgroup (Fig. 5B). LMC1 was highest in pulmonary carcinoids, LMC3 and LMC10 in appendiceal NEN, LMC4 in pancreatic NEN, LMC5 in ileal NEN, LMC6 in colorectal NEN, LMC7 in pulmonary NEC and LMC9 in Merkel cell carcinoma. No clear organ-site specificity could be

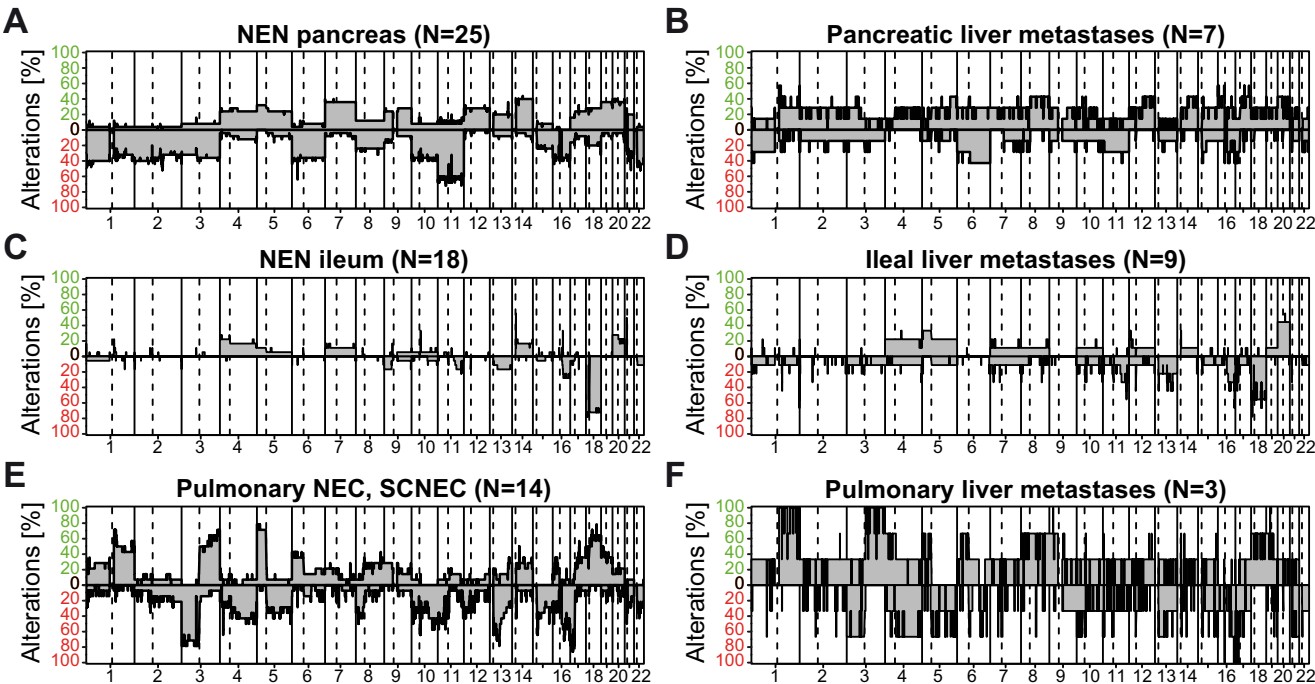

**Fig. 4 | Copy number profiles show distinct genetic alterations in NEN of different organ sites and indicate heterogeneous alterations in hepatic NEN.**
**A** NEN of the pancreas ($N = 25$), **B** liver metastasis of pancreatic NEN (N = 7), **C** NEN of the ileum ($N = 18$), **D** liver metastasis of ileal NEN ($N = 9$), **E** pulmonary small-cell

NEC (SCNEC) ($N = 14$) and **F** liver metastasis of pulmonary NEN ($N = 3$). The relative frequency of observed gains (green) and losses (red) are depicted above and below the horizontal line, respectively.

observed for LMC8. Consistent with the varying origins of the NEN liver metastases, this group of NEN showed high LMC4 (pancreatic NEN), high LMC5 (ileal NEN), LMC6 (colorectal NEN) and LMC7 (pulmonary NEC; Fig. 5B). Next, the LMC were used to train Random Forest machine learning algorithms for classification of NEN subgroups (see Materials and Methods for details). This resulted in excellent prediction accuracies with a mean area under the curve (AUC) of 0.99 for Merkel cell carcinoma, ileal NEN and pulmonary carcinoid. The mean AUC for appendiceal NEN, pulmonary NEC, pancreatic NEN, colorectal NEN and gastric/duodenal NEN reached 0.98, 0.97, 0.96, 0.93 and 0.92, respectively (Supplementary Fig. 10). Similarly, our prediction algorithm correctly classified 19 of 22 (86.4%) NEN liver metastases, whereby 2 of the 3 misclassified cases were classified to other locations within the gastrointestinal system, confirming the feasibility of our classification approach using liver biopsy material (Supplementary Data 6). To compare our LMC-based Random Forest prediction algorithm with another machine learning algorithm, we applied the widely used XGBoost algorithm. LMC-Random Forest and XGBoost yielded comparable prediction performance on the NEN reference groups; however, only 17 of 22 (77.3%) NEN liver metastases were classified correctly (Supplementary Fig. 11). Therefore, we chose the Random Forest algorithm for further analyses. The Random Forest prediction algorithm predicted an organ of origin outside the liver for all patients with hepatic NEN (Table 4). In detail, two hepatic NEN were predicted to be ileal NEN, five colorectal NEN, five pancreatic NEN, two gastric/duodenal NEN, and eight pulmonary NEC. However, the level of confidence was not equally high for all NEN organ sides. Especially the two samples which were predicted to be gastric/duodenal NEN had only slightly lower prediction scores for colorectal NEN and likewise one NEN predicted as colorectal NEN had a slightly lower prediction score for gastric/duodenal NEN suggesting that the algorithm is less powerful in distinguishing between gastric/duodenal and colorectal NEN (Table 4). This finding is consistent with the low clustering of gastric/duodenal NEN, liver metastases of gastric/duodenal NEN and hepatic NEN predicted to be gastric/duodenal in the t-SNE plot (Fig. 5C). In

contrast, the hepatic NEN which were predicted by the Random Forest algorithm to be of pancreatic origin clustered together with the pancreatic NEN of the reference group and liver metastasis of pancreatic NEN indicating high similarity (Fig. 5D). Similarly, hepatic NEN predicted to be of pulmonary NEC or predicted to be of ileal origin fell into the cluster of pulmonary NEC and liver metastasis of pulmonary NEC or into the cluster of primary ileal NEN and ileal metastases (Fig. 5E, F). Consistently, the genomic profiles of the hepatic NEN predicted to be ileal NEN, pulmonary NEC or pancreatic NEN each resembled the profiles of the respective reference cohort (Supplementary Fig. 12). Interestingly, almost all hepatic NEN without known primary were predicted to originate from lung NEC, pancreatic NET or small bowel NET, which all exhibit foregut embryological origin. This suggested that a foregut DNA methylation pattern is present in most hepatic NEN without known primary. Thus, tracing origin sites of NEN is a feasible task for DNA methylation analysis. However, the existence of a distinct, separable entity of primary hepatic NEN cannot be proven according to comprehensive and systematic DNA methylation analyses of intrahepatic and extrahepatic NEN in this study.

For independent validation, we performed DNA methylation analysis of an independent hepatic NEN cohort from Beaujon University Hospital (Paris, France). When compiling this cohort, any patients with unclear imaging or with NEC were excluded (Fig. 6A)[25]. This resulted in a total of 14 patients with hepatic NEN including one patient with two hepatic NEN tumors. Therefore, 15 hepatic NEN of 14 patients were analyzed by DNA methylation profiling. Similar to those hepatic NEN from the Heidelberg cohort, the hepatic NEN without known primary from the Beaujon cohort did not form a distinct cluster in t-SNE analysis. Instead, they overlapped and clustered with different extrahepatic NEN subtypes (Fig. 6B). To further specify the potential NEN organ site of the hepatic NEN from the Beaujon cohort, we again applied the Random Forest prediction algorithm, as already used for the Heidelberg cohort. Thereby, two hepatic NEN were predicted to be pulmonary carcinoids, two samples which were derived from the same patient were both predicted to be appendiceal NEN, two samples were

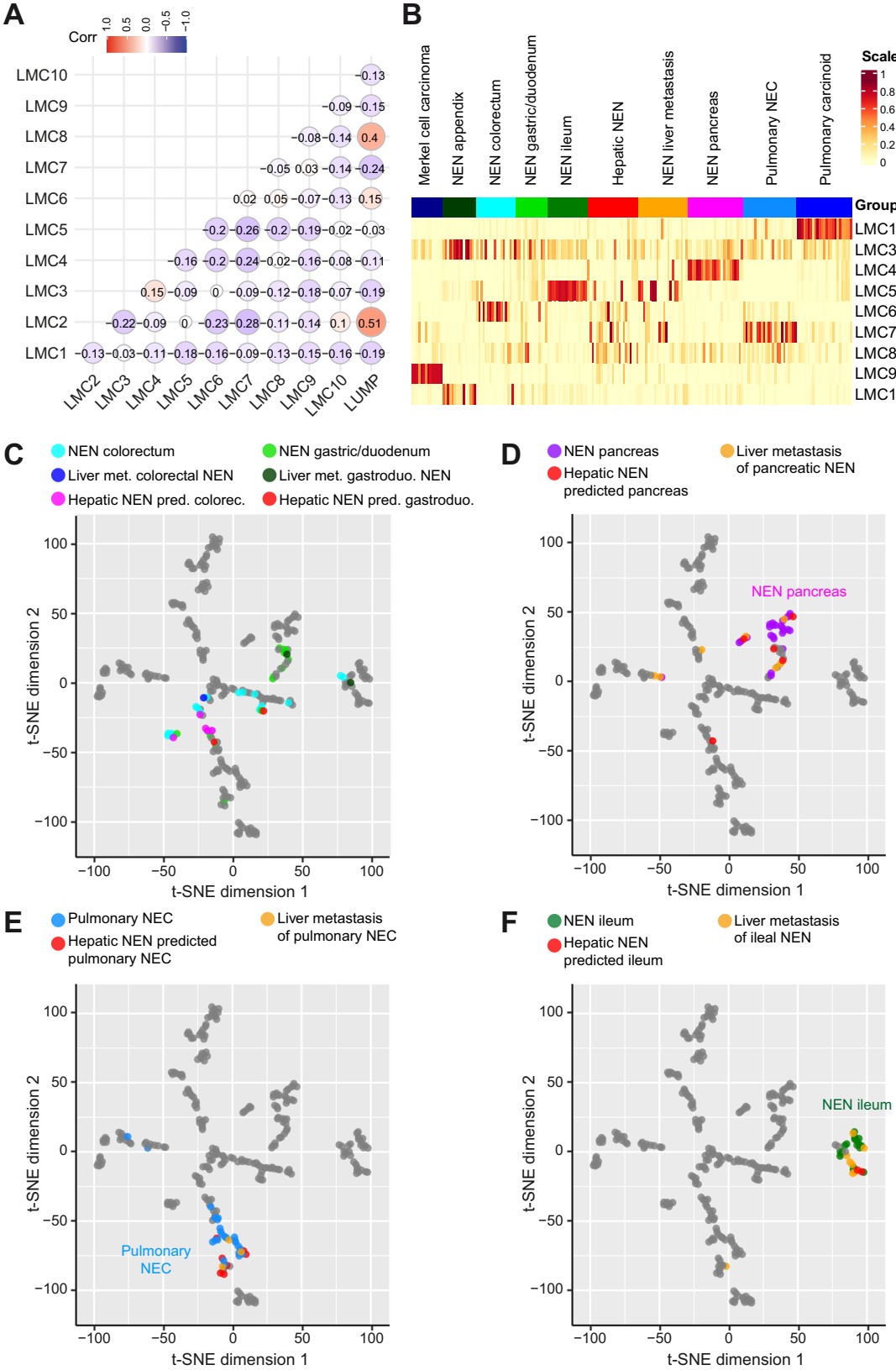

predicted to be ileal NEN and nine hepatic NEN were predicted as gastric/duodenal NEN (Supplementary Data 7). Consistently, the hepatic NEN predicted to originate from the lung and the gastric/duodenal and colorectal NEN clustered with the respective NEN reference groups (Fig. 6C, D). Thus, we were able to confirm that for most hepatic NEN an extrahepatic origin is likely.

## Discussion

Patients with NEN continue to pose significant challenges for both clinicians and pathologists. Uncertain diagnoses can influence therapeutic decisions and may lead to suboptimal treatment. From a diagnostic perspective, accurate tumor classification, including precise subclassification, remains a complex task. In the metastatic

**Fig. 5 | Decomposition of DNA methylation profiles assigns most hepatic NEN to extrahepatic origins. A** Correlation of latent methylation components (LMC) with each other and with Leukocytes Unmethylation for Purity (LUMP) indicating lymphocyte infiltration. **B** Supervised heatmap of LMC enrichment in Merkel cell carcinomas ($N=14$), appendiceal NEN ($N=15$), colorectal NEN ($N=18$), gastric/duodenal NEN (N=14), ileal NEN ($N=18$), hepatic NEN without known primary tumor ($N=22$), NEN liver metastases of known primary ($N=22$), pancreatic NEN ($N=25$), pulmonary NEC ($N=24$) and pulmonary carcinoids ($N=25$). **C**–**F** t-SNE plots of the DNA methylation profiles of CCA, HCC and a total of 197 NEN samples from different organ sites. **C** Colorectal NEN ($N=18$), gastric/duodenal NEN ($N=14$),

liver metastases of colorectal NEN ($N=1$), liver metastases of gastro/duodenal NEN ($N=1$), hepatic NEN predicted to be colorectal NEN ($N=3$) and hepatic NEN predicted to be gastric/duodenal NEN ($N=3$) are highlighted in color. **D** NEN of the pancreas ($N=25$), liver metastases of pancreatic NEN ($N=7$) and hepatic NEN predicted to be pancreatic NEN ($N=4$) are highlighted in color. **E** Pulmonary NEC ($N=24$), liver metastases of pulmonary NEC ($N=3$) and hepatic NEN predicted to be pulmonary NEC ($N=10$) are highlighted in color. **F** Ileal NEN ($N=18$), liver metastases of ileal NEN ($N=7$) and hepatic NEN predicted to be ileal NEN ($N=2$) are highlighted in color. LMC latent methylation components, LUMP Leukocytes Unmethylation for Purity.

setting, identifying the primary tumor using conventional histopathological and immunohistochemical tools is often not feasible.

In this study, we present a comprehensive morphomolecular analysis of NEN originating from various anatomical sites and encompassing multiple histological subtypes and tumor grades. This analysis provides a broad overview of the epigenetic landscape of NEN. Our findings demonstrate that epigenetic profiling, particularly methylome analysis, holds strong potential for the subclassification of NEN and for identifying their origin for most anatomical locations. Moreover, methylation pattern analysis enables differentiation of NEN not only by site of origin and tumor grade, but also allows clear distinction of NEN within the liver from primary liver cancers, such as HCC and CCA. Consequently, DNA methylation emerges as a powerful tool not only for confirming the diagnosis of NEN but also for tracing the tumor's primary site, thereby potentially improving therapeutic strategies.

A recent study demonstrated the potential of DNA methylation analysis in predicting the origin of NEN[33]. Remarkably, one case of hepatic NEN with unknown primary was predicted an ileal NEN as the primary tumor using DNA methylation analysis[33]. However, this intriguing phenomenon has not been systematically followed up in subsequent studies.

In light of the ongoing debate surrounding the existence and prevalence of primary hepatic NEN, our findings provide no evidence of a distinct organ-specific methylation signature for tumors initially diagnosed as primary hepatic NEN. In contrast, analysis of a subcohort of paired primary and metastatic samples revealed a high degree of methylation pattern concordance between liver metastases and their corresponding primaries. This underscores the diagnostic potential of DNA methylation analysis in identifying both the primary tumor site and the cell-of-origin, while simultaneously questioning the existence of primary hepatic NEN as a distinct entity. While we acknowledge the possibility of rare cases of true primary hepatic NEN, our data suggest that many tumors labeled as such may in fact represent metastases from small, undetected, or regressed primary tumors at extrahepatic sites – a phenomenon well-documented in other malignancies such as malignant melanoma[34–36].

In cases of genuine primary hepatic NEN, our findings suggest heterogenous cellular origins and a broad spectrum of tumor cell differentiation. Furthermore, the observed similarity in DNA methylation profiles and immunohistochemical markers between hepatic and extrahepatic NEN supports the rationale for applying therapeutic regimens analogous to those used for extrahepatic NEN. In addition, we were able to show that primary hepatic NEN and NEN metastases exhibited similar clinical courses and that overall survival depended in particular on the tumor subtype and grading irrespective of applied therapeutic regimens (Table 1; Supplementary Fig. 1). This is consistent with previous findings and may guide clinical decision-making for this rare tumor entity[37,38]. Unlike pancreatic, bronchopulmonal and small bowel NEN, which are characterized by well-defined clinical and molecular profiles, the genomic landscape of primary hepatic NEN has only recently begun to be investigated[25]. Interestingly, our results are consistent with prior studies identifying a foregut-like molecular signature in 'metastatic NET of unknown primary'[25]. However,

comprehensive DNA methylation analyses have not been performed in hepatic NEN so far.

Using both supervised and unsupervised DNA methylation analyses, nearly all samples of our cohort of hepatic NEN without known primary colocalized with lung NEC, pancreatic NET or small bowel NET, which all exhibit foregut embryological origin, but the discovery cohort of hepatic NEN (Heidelberg cohort) showed clear heterogeneity. Consistently, a specific methylation profile in presumed primary hepatic NEN as seen in pancreatic NET could not be detected[25,39]. These observations were independently validated in a separate cohort from Paris (Beaujon cohort), which not only confirmed the absence of a liver-specific methylation cluster but also supported the robustness of our DNA methylation pattern-based NEN classifier.

The challenges in detecting small submucosal NEN in anatomically less accessible regions may explain negative staging results. However, the lack of precise guidelines and standalone markers for classifying hepatic NEN complicates the differentiation between primary hepatic NEN and metastasis from an undetected extrahepatic origin[16,40]. Widely accepted, the liver is a very frequent site of metastasis of NEN primaries located at various extrahepatic sites, in particular the gastrointestinal tract. In contrast, primary hepatic NEN are believed to be exceedingly rare and only a few reports of primary hepatic NEN exist to date[41–44].

NEN throughout the human body are thought to originate from diffuse neuroendocrine networks present in most organ sites, such as the small and large intestines or the lungs. With regard to the hepatobiliary situation, this diffuse network of neuroendocrine cells is detectable in the hilar region of the liver and along the extrahepatic biliary tree, and subsequently NEN and MiNEN are known to occur rarely in the perihilar and extrahepatic biliary tract. However, within the normal liver parenchyma, there is no detectable diffuse network of neuroendocrine cells, a finding that complicates the concept of primary hepatic NEN per se. Another argument against the concept of primary hepatic NEN is the absence of a distinct t-SNE cluster of hepatic NEN without known primary, together with the observation that the majority of metastatic NEN analyzed in paired samples - i.e. NEN originating from the primary anatomical site and corresponding metastases - exhibited a foregut-like phenotype, similar to that of hepatic NEN without known primary. As we cannot confirm the presence of genuine primary hepatic NEN at the molecular level, it seems more likely that most NEN within the liver and without a primary extrahepatic NEN detectable account for metastases of foregut- or midgut-type NEN which were clinically not detectable.

Copy number alterations (CNA) represent unique signatures specific to organ sites including NEN[29,45,46]. In this study, CNA profiles of hepatic NEN predicted to originate from pancreatic NEN, ileal NEN or pulmonary NEC were similar to the reference CNA profiles supporting the DNA methylation-based prediction. Similar to a recent study, the ileal NEN and NEN predicted to be ileal exhibited a high frequency of chromosome 18 deletion[45]. In addition, our CNA profiles of appendiceal NET revealed only few and mostly minor alterations, which might result in the clinically very benign behavior of appendiceal NET[46].

**Table 4 | Predicted origin of hepatic NEN (N = 22)**

| Patient | Clus-ter | Grad-ing | Inter-vention | Prediction | Prediction score | | | | | | | |
|---|---|---|---|---|---|---|---|---|---|---|---|---|
| | | | | | Merkel cell carcinoma | Appendiceal NEN | Colorectal NEN | Gastric/ duode-nal NEN | Ileal NEN | Pancreatic NEN | Pulmonary NEC | Pulmonary carcinoid |
| Patient #038 | 2 | NET G2 | Bx | NEN ileum | 0.022 | 0.055 | 0.040 | 0.065 | **0.627** | 0.123 | 0.015 | 0.054 |
| Patient #018 | 2 | NET G2 | Bx | NEN ileum | 0.028 | 0.073 | 0.073 | 0.119 | **0.427** | 0.166 | 0.019 | 0.095 |
| Patient #035 | 1 | SCNEC | Bx | NEN colorectal | 0.073 | 0.041 | **0.507** | 0.097 | 0.020 | 0.035 | 0.201 | 0.027 |
| Patient #032 | 1 | NET G3 | Bx | NEN colorectal | 0.043 | 0.034 | **0.375** | 0.168 | 0.051 | 0.115 | 0.171 | 0.043 |
| Patient #036 | 1 | LCNEC | Bx | NEN colorectal | 0.053 | 0.041 | **0.331** | 0.174 | 0.053 | 0.091 | 0.215 | 0.041 |
| Patient #037 | 1 | LCNEC | Bx | NEN colorectal | 0.043 | 0.055 | **0.305** | 0.219 | 0.099 | 0.094 | 0.150 | 0.034 |
| Patient #030 | 1 | NET G3 | Rx | NEN colorectal | 0.038 | 0.094 | **0.220** | 0.199 | 0.141 | 0.135 | 0.125 | 0.048 |
| Patient #029 | 1 | SCNEC | Rx | Pulmonary NEC | 0.087 | 0.035 | 0.083 | 0.067 | 0.006 | 0.025 | **0.666** | 0.032 |
| Patient #017 | 1 | LCNEC | Bx | Pulmonary NEC | 0.043 | 0.029 | 0.178 | 0.077 | 0.008 | 0.028 | **0.600** | 0.037 |
| Patient #026 | 1 | LCNEC | Bx | Pulmonary NEC | 0.101 | 0.045 | 0.172 | 0.086 | 0.021 | 0.041 | **0.493** | 0.042 |
| Patient #025 | 1 | LCNEC | Bx | Pulmonary NEC | 0.187 | 0.050 | 0.161 | 0.109 | 0.010 | 0.035 | **0.414** | 0.034 |
| Patient #028 | 1 | LCNEC | Bx | Pulmonary NEC | 0.039 | 0.061 | 0.239 | 0.125 | 0.019 | 0.069 | **0.395** | 0.053 |
| Patient #016 | 1 | NET G2 | Bx | Pulmonary NEC | 0.163 | 0.042 | 0.133 | 0.123 | 0.040 | 0.065 | **0.366** | 0.068 |
| Patient #031 | 1 | LCNEC | Bx | Pulmonary NEC | 0.231 | 0.081 | 0.179 | 0.107 | 0.013 | 0.035 | **0.321** | 0.033 |
| Patient #034 | 1 | LCNEC | Rx | Pulmonary NEC | 0.204 | 0.069 | 0.141 | 0.158 | 0.016 | 0.061 | **0.290** | 0.060 |
| Patient #013 | 2 | NET G2 | Rx | NEN pancreas | 0.018 | 0.040 | 0.059 | 0.232 | 0.080 | **0.395** | 0.050 | 0.127 |
| Patient #033 | 2 | NET G1 | Rx | NEN pancreas | 0.037 | 0.046 | 0.067 | 0.179 | 0.158 | **0.343** | 0.061 | 0.109 |
| Patient #015 | 2 | NET G2 | Bx | NEN pancreas | 0.035 | 0.033 | 0.052 | 0.135 | 0.136 | **0.336** | 0.029 | 0.245 |
| Patient #039 | 2 | NET G1 | Rx | NEN pancreas | 0.021 | 0.055 | 0.045 | 0.300 | 0.088 | **0.305** | 0.057 | 0.131 |
| Patient #021 | 1 | LCNEC | Bx | NEN pancreas | 0.089 | 0.060 | 0.127 | 0.187 | 0.163 | **0.207** | 0.094 | 0.073 |
| Patient #020 | 2 | NET G1 | Rx | NEN gastric/ duodenal | 0.020 | 0.146 | 0.223 | **0.231** | 0.081 | 0.164 | 0.079 | 0.056 |
| Patient #014 | 1 | NET G3 | Bx | NEN gastric/ duodenal | 0.070 | 0.040 | 0.138 | **0.227** | 0.086 | 0.158 | 0.215 | 0.066 |

Prediction scores indicating probability of organ site are indicated as proportional probability.

Bx Biopsy, Rx Resection, NEC Neuroendocrine carcinoma, NET Neuroendocrine tumor, LCNEC Large-cell neuroendocrine carcinoma, SCNEC Small-cell neuroendocrine carcinoma, Bold values indicate the highest prediction score within each sample.

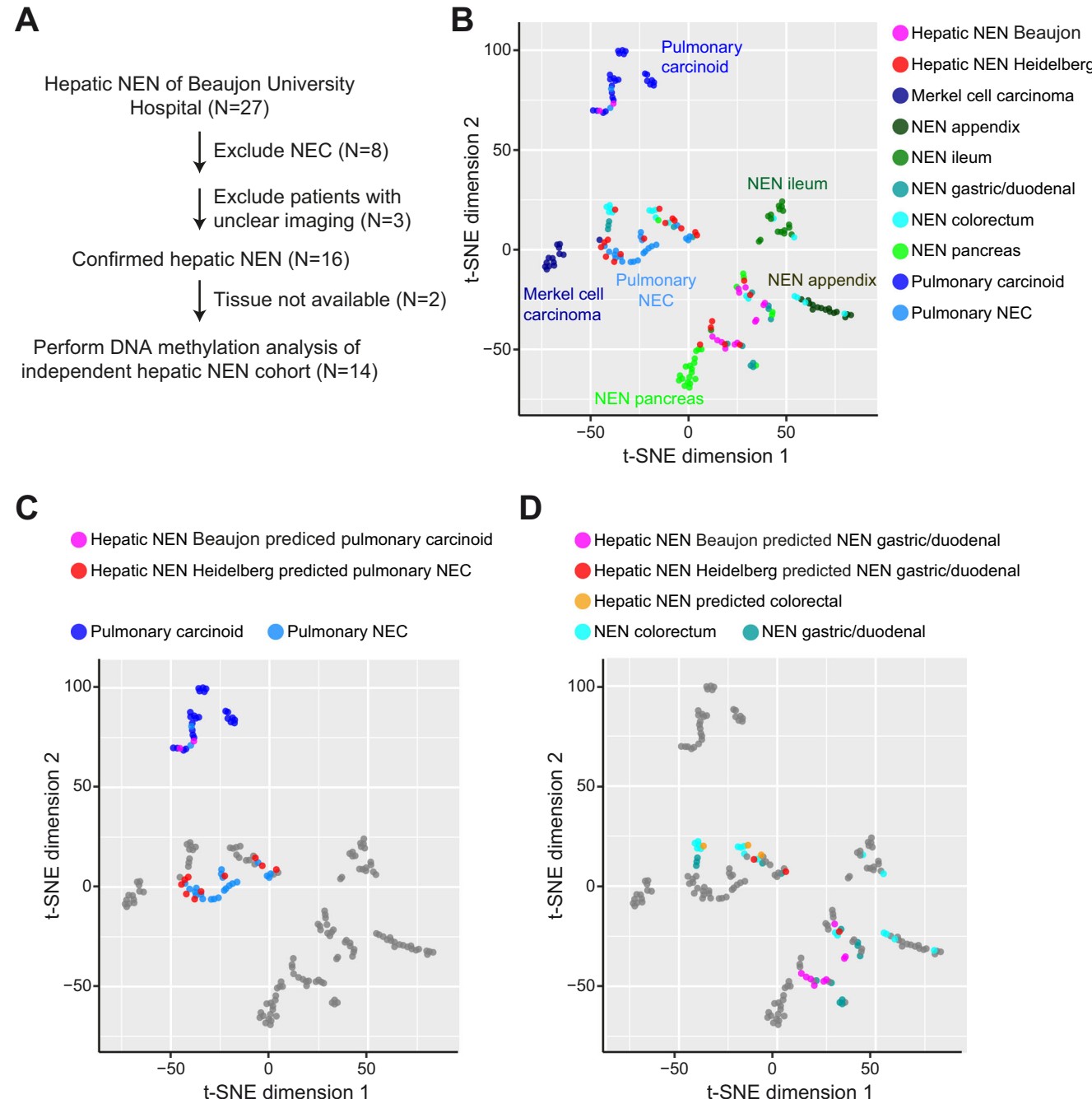

**Fig. 6 | Hepatic NEN of the independent validation cohort from Beaujon University Hospital. A** Overview of patient selection process for the independent validation cohort from Beaujon University Hospital (Paris, France). **B**–**D** t-SNE plots of the DNA methylation profiles of the hepatic NEN from Beaujon ($N = 15$) and Heidelberg University Hospitals ($N = 22$) and of the NEN reference cohort ($N = 153$) including Merkel cell carcinoma ($N = 14$ of 11 patients), appendiceal NEN ($N = 15$), ileal NEN ($N = 18$), gastric/duodenal ($N = 14$), colorectal NEN ($N = 18$), pancreatic NEN

($N = 25$), pulmonary carcinoids ($N = 25$) and pulmonary NEC ($N = 24$). (**C**). Hepatic NEN from Beaujon predicted as pulmonary carcinoids and hepatic NEN from Heidelberg predicted as pulmonary NEC, together with pulmonary carcinoids ($N = 25$) and pulmonary NEC ($N = 24$) of the reference cohort are highlighted in color. **D** Hepatic NEN from Beaujon and Heidelberg University Hospitals predicted as gastric/duodenal NEN or as colorectal as well as the reference groups of colorectal NEN and gastric/duodenal NEN are highlighted in color.

A limitation of this study is that for some hepatic NEN, the prediction scores of two NEN subtypes were similar (especially between colorectal and gastric/duodenal NEN) and therefore, the exact origin could not always be predicted with high confidence. In addition, the number of gastric, duodenal and colonic NEN was relatively low in the analyzed cohort. This number may not be sufficient to accurately discriminate between primary tumors of these locations using DNA methylation patterns alone. Therefore, follow-up studies should address this issue by analyzing larger groups of these NEN subtypes, improving the ability to

predict the primary site of NEN using DNA methylation patterns. Further studies should aim to correlate whole exome genetic and global epigenetic alterations employing a well-characterized and comprehensive multi-institutional cohort of different NEN. This may provide additional information that will not only allow us to better diagnose and subtype these tumors, but to also provide a more accurate prognosis and identify putative therapeutic targets for NEN patients.

In conclusion, by analyzing the DNA methylation profiles of a large and comprehensive cohort of NEN from different anatomical

locations, we were able to i) detect differences in the DNA methylation profile of NEN from different locations, ii) present a tool for tracing the origin of NEN for most anatomical locations, and iii) address open questions in the classification of NEN of unknown primary. In this regard, this study may improve clinical decision making by supporting clinicopathological diagnostic algorithms for tracking NEN of unknown primary and improve our understanding of the complex situation of NEN within the liver in general.

## Methods

### Study population and histomorphological subclassification

The discovery cohort of this study included 197 tissue samples of 185 patients with NEN localized to the liver, pancreas, stomach, duodenum, ileum, appendix, colorectum, lung or skin from Heidelberg University Hospital. Clinicopathologic data of patients are presented separately for intrahepatic and extrahepatic tumors (Tables 1 and 2). Of 9 patients with liver metastases and corresponding extrahepatic primary tumors, the extrahepatic tumors were also included to the extrahepatic NEN study group according to their original locations. In detail, hepatic NEN without known primary tumor ($N = 22$), hepatic metastases of known primary NEN ($N = 22$), of which 9 were paired samples, gastric/duodenal NEN ($N = 14$), appendiceal NEN ($N = 15$), ileal NEN ($N = 18$), colorectal NEN ($N = 18$), pancreatic NEN ($N = 25$), pulmonary NEC ($N = 24$), pulmonary carcinoids ($N = 25$) and Merkel cell carcinomas ($N = 14$ tissues of 11 patients) were included. Therefore, the study comprised 197 NEN tissues of 185 patients.

NEN were diagnosed according to the current histomorphological and immunohistochemical criteria provided by WHO[15]. In this context, NEN were classified as neuroendocrine tumors (NET) of different tumor grade (G1-3), neuroendocrine carcinomas (NEC) and mixed neuroendocrine/non-neuroendocrine neoplasms (MiNEN). Immunohistochemical positivity for SYP and CHGA was used to confirm neuroendocrine differentiation. MiNEN were defined as mixed epithelial neoplasms in which a neuroendocrine component is combined with a non-neuroendocrine component, each of which is morphologically and immunohistochemically recognizable as a discrete component and constitutes at least 30% of the neoplasm each[15]. All patients with hepatic NEN included in this study were discussed in a multidisciplinary tumor board.

The independent validation cohort (Beaujon cohort) included 15 hepatic NEN of 14 patients treated and diagnosed at Beaujon University Hospital, Paris, France. Patients of this cohort were already described in detail by de Mestier et al. [25]. Patients with neuroendocrine carcinoma were not included in this validation cohort.

In addition, epigenetic profiles of NEN were compared to CCA, HCC, non-neoplastic bile duct and non-neoplastic liver tissues. Therefore, we retrieved all paired DNA methylation data sets of HCC patients available at TCGA. A total of 50 paired DNA methylation data sets of HCC and non-neoplastic normal liver tissues were available from the TCGA-LIHC database (https://www.cancer.gov/tcga). For CCA and non-neoplastic bile duct, our previous DNA methylation data was used. This comprised 27 invasive CCA and 50 non-neoplastic normal bile ducts (https://www.ncbi.nlm.nih.gov/geo/; GSE156299).

### Immunohistochemical analysis

For immunohistochemistry, 3 μm sections were cut, deparaffinized and rehydrated. Ultra CC1 (Cell Conditioning Solution, Ventana Medical Systems, Tucson, AZ, USA) was used for heat-induced epitope retrieval. After blocking of endogenous peroxidase, slides were incubated with primary antibodies. Detailed information regarding the antibodies used is provided in Supplementary Data 8. Biotin-free OptiView DAB IHC Detection Kit (Ventana Medical Systems) including OptiView Universal Linker, OptiView HRP Multimer and DAB-Chromogen was used. Finally, the slides were counterstained with hematoxylin.

The scoring of immunohistochemical results was performed serving the diagnostic purpose of each immunohistochemical marker. According to the WHO classification, Ki67 analysis was performed by counting at least 500 tumor cells in hotspot areas, thereby determining the percentage of Ki67 positive tumor cells. For p53 immunohistochemical analysis, the established rules for discriminating wild type and mutational type were applied[47]. For RB1, the complete negativity in tumor cells was regarded as RB1 loss. Nuclear staining in at least 10% of tumor cells was defined as positive result for ARX and PDX1. All other immunohistochemical analyses were scored using a four-tiered algorithm. Positive: homogeneously positive in tumor cells, positive/negative: positive in the majority of tumor cells (>50%), negative/positive: positive in the minority of tumor cells (<50%), negative: completely negative in the tumor cells. Positivity for panCK was determined by positive staining of antibodies specific for CK7, CK20 or panCK. The non-neoplastic cells on the whole slide image were used as a control for all performed immunohistochemical analyses.

### Genomic DNA isolation

All NEN samples were pathologically assessed to identify regions with the highest tumor content. In all cases, tumor cell content exceeded 50%, and tumor cell purity was further confirmed using LUMP[48] analysis based on the DNA methylation profiles (Supplementary Fig. 13). Genomic DNA was extracted from formalin-fixed paraffin-embedded (FFPE) samples using the AllPrep DNA/RNA FFPE Kit (Qiagen, Hilden, Germany) according to the manufacturer's instructions with the following modifications: After addition of xylene, samples were incubated at 56 °C for 2 min, followed by two ethanol washes. The initial proteinase K digestion was performed with 20 μl at 56 °C for 30 min. DNA was eluted twice with 30 μl of $H_2O$.

### DNA methylation analysis using Infinium MethylationEPIC array data processing

DNA methylation profiles were determined by the Genomics and Proteomics Core Facility (DKFZ Heidelberg) using the Infinium MethylationEPIC BeadChip assay and the Infinium MethylationEPIC v2.0 assay for the independent validation set from Beaujon University Hospital (Illumina, San Diego, CA, USA). The assay determined DNA methylation levels and allowed for the quantitative measurement of CNA. FFPE tissue-derived genomic DNA was treated with bisulfite using the EZ DNA Methylation kit (D5002, Zymo Research, Irvine, CA, USA). Infinium MethylationEPIC and MethylationEPIC v2.0 arrays were performed according to the manufacturer's instructions and scanned on an Illumina HiScan. The assay determined DNA methylation levels at >850,000 CpG sites and allowed for the quantitative measurement of CNA.

Illumina EPIC and 450 K methylation array samples were merged into a single dataset using their common probes according the manufactures annotations using the minfi package. Each sample then underwent individual normalization, which included a background correction and a dye-bias correction (scaling normalization control probe intensities). Following this, a batch correction was applied to the log2-transformed intensity values using univariable linear models with the limma package. This step adjusted for variations of tissue type (FFPE or frozen) and array type (450 K or EPIC), with methylated and unmethylated signals corrected separately. Finally, beta-values were calculated from the re-transformed intensities, incorporating an offset of 100. The CpG probes are filtered according the filter criteria described in Patel et al.[49].

The t-SNE plots were computed via the R package Rtsne using the 10,000 most variable CpG sites according to standard deviation, 3,000 iterations and a perplexity value of 10. Copy-number variations were calculated from the IDAT files using the R/Bioconductor package conumee including an additional baseline correction (https://github.com/dstichel/conumee and http://bioconductor.org/packages/

release/bioc/html/conumee.html). As HCC and paired non-neoplastic liver DNA methylation profiles of TCGA-LIHC were performed using Illumina Human Methylation 450, data analysis for plots which included these samples was restricted to the overlapping CpG sites, as described above.

SNP analysis was performed on SNP sites included in the Illumina EPIC and was restricted to sites shared across all samples avoiding missing-data bias. For each pair of samples i and j, we computed the SNP distance

$$d_{ij} = \frac{\sum_{k=1}^{M} 1\{g_{i,k} \neq g_{i,k}\}}{M}$$

where M is the total number of shared SNP loci and $g_{i,k}$ the genotype (homozygous reference, heterozygous, or homozygous alternate) of sample i at locus k. Hierarchical clustering and the heatmap depicting the computed SNP distances were generated using the R package ComplexHeatmap[50].

For CNA analysis, we used the R package conumee as reported previously (http://bioconductor.org/packages/conumee/)[27,28,51]. Every interrogated CpG of the Illumina DNA methylation microarrays is represented by two probes on the array. One of these probes detects the methylated and the other one the unmethylated CpG. For the calculation of CNA, the methylated and unmethylated signal intensities are added together and a ratio is formed against healthy reference samples that have a flat genome. This copy-number ratio is then plotted in a graph according to chromosomal location.

For further analyses, the DNA methylation array data were processed as described previously[27]. Briefly, the Infinium MethylationEPIC arrays were preprocessed using R 4.1 and RnBeads[52,53] version 2.10.0. Probes overlapping or close to common SNPs ($N = 17,371$) and cross-reactive probes (N = 43,463), 93,216 probes of low quality based on the Greedycut algorithm as implemented in RnBeads were removed. In addition, probes located on sex chromosomes were not used ($N = 13,420$). The data was normalized with the SWAN normalization algorithm, as implemented in the minfi[54] package combined with the methylumi package's noob background correction.

We performed a deconvolution of the methylation data using MeDeCom[31,32]. Included sites were selected based on the following criteria: using differential methylation analysis by limma[55] comparing each cancer site with the rest of the samples, we identified the most different (by methylation difference) 500 probes by site. We then combined them with the top 5000 most variable CpG sites. The final model was based on 12,291 unique sites. After optimization, the MeDeCom analysis identified 10 latent methylation components. Based on Pearson correlation analysis, we have identified LMC2 to be associated with the normal cell content of the tumors, therefore LMC2 was not included on the visualization of the components.

A Random Forest classifier was developed to predict the tissue of origin of NEN of different organ sites. First, we intersected the 12,291 unique CpG sites of our discovery cohort with those in the validation cohort, yielding 9604 common CpGs. From these, we retained the 5000 most variable sites for model training. To eliminate confounding by normal-cell contamination, we adjusted each CpG's methylation level for its association with the LMC2 methylation component: for every CpG, we fitted a linear regression against LMC2 proportion, and −if that association was significant−we used the residuals from this model in place of the raw methylation values. The Random Forest classifier was trained on the selected sites using the scikit-learn Python (3.8) package with the following parameters: n_estimators=1500, random_state=3, max_features = "sqrt", criterion = "gini", oob_score=True, n_jobs=10, max_depth=12. The model was trained on the NEN samples, excluding known metastases and liver CUPs. Stratified 3-fold cross-validation was performed to evaluate the classifier's accuracy. The out-of-bag (OOB) score was computed for model performance assessment.

Feature importance was assessed by calculating the importance values from the model. Receiver Operating Characteristic (ROC) curves were plotted for each class using stratified 3-fold cross-validation and the area under the ROC curve (AUC) was computed. For each sample, the highest value determined the predicted organ site/NEN subtype. The trained model was further evaluated by predicting the grouping of the NEN liver metastases subset.

For comparison, the XGBoost model using the R package xgboost (1.7.11.1) was applied to the 9604 common CpG sites. From this model, we extracted the 404 CpG sites with the highest importance scores as our most informative features. These 404 CpGs were then used to train a Random Forest classifier following the same procedure described above: We fit the Random Forest model on the selected features and evaluated its performance in predicting the tissue of origin of neuroendocrine neoplasms across different organ sites.

### Statistics & reproducibility
This is a retrospective study using clinicopathologically characterized cohorts of patients with NEN. No statistical method was used to pre-determine sample size. No data were excluded from the analyses. The experiments were not randomized. The investigators were not blinded to allocation during experiments and outcome assessment. MeDeCom analysis and Random Forest classifier development were independently performed by R.T., Y.Z. and D.T. and all findings were replicated. Stratified 3-fold cross-validation was performed to evaluate the classifier's accuracy. Statistical analysis and visualization were performed using the computing environment R and GraphPad Prism 8. All reported p-values were two-sided and $p < 0.05$ was considered statistically significant.

### Ethics statement
All research was conducted in accordance with relevant ethical guidelines. All tissue samples of the Heidelberg cohort were provided by the Tissue Bank of the National Center for Tumor Diseases (NCT, Heidelberg, Germany) in accordance with the regulations of the NCT Tissue Bank. All patients included underwent surgical procedures at the University Hospital Heidelberg between 2008 and 2019. The study protocol conformed to the ethical guidelines of the 1975 Declaration of Helsinki and was approved by the ethics committee of the University of Heidelberg (S-206/2005, S-207/2005, S-519/2019 and S-043/2024). Each tumor sample was histologically confirmed by at least two experienced consultant pathologists. Analysis of the tissue samples of the Beaujon cohort was conducted in accordance with the STROBE/SRQR guidelines and Helsinki declaration, following Institutional Review Board approval (CEERB Paris Nord, IRB 00006477-15-073). All patients received information and their non-opposition was recorded. Patients were not involved in the design of the study.

### Reporting summary
Further information on research design is available in the Nature Portfolio Reporting Summary linked to this article.

## Data availability
All relevant data is included in the manuscript, supplementary material or made publicly available. The DNA methylation data generated in this study by Infinium MethylationEPIC BeadChip have been deposited in the Gene Expression Omnibus (GEO) database and are available at the under accession GSE253176 for the NEN cohort form Heidelberg and under accession GSE298677 for the validation cohort form Beaujon University Hospital, Paris.

## Code availability
The code for DNA methylation analysis is available at Github [https://github.com/Multi-omics-Data-Science-LIH/NEN-methylation], [https://doi.org/10.5281/zenodo.17193743].

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

## Acknowledgements

We are grateful to Veronika Geißler, Nina Wilhelm and Carolin Kerber (Tissue Bank of the National Center for Tumor Diseases (NCT) Heidelberg, Germany and Institute of Pathology Heidelberg) and Angelika Fraas (Institute of Pathology Heidelberg) for excellent technical assistance. Tissue samples were provided by the tissue bank of the National Center for Tumor Diseases (NCT; Heidelberg, Germany) in accordance with the regulations of the tissue bank and with approval of the Ethics Committee of Heidelberg University (S-206/2005, S-207/2005, S-519/2019 and S-043/2024). We would like to thank the NCT Cancer Registry for providing patient survival data. This work was supported by Deutsche Forschungsgemeinschaft (DFG, German Research Foundation) – project-ID 314905040 – SFB/TRR 209 Liver Cancer B01, Project-ID 469332207 and project-ID 493697503 to SR. It was also in part supported by funds from German Cancer Aid (Deutsche Krebshilfe, project no. 70113922) to SR. This work was supported by an Illumina research grant to AvD. We thank the Genomics and Proteomics Core Facility (GPCF) of DKFZ for performing the DNA methylation profiling. For publication fee, we acknowledge financial support by Heidelberg University.

## Author contributions

B.G. initiated the study; B.G. and S.R. designed the research project; B.G, A.C., T.A., M.V., S.S., L.M., J.C., D.C., A.B. and L.A. provided tissue samples and/or clinicopathological data; B.G., A.C and S.R. performed research experiments; B.G., A.C, R.T, Y.Z, D.T., T.A., D.S., E.B., A.B., J.J., M.V., S.S., P.V.N., A.P., Av.D. and S.R. analyzed data or tissue samples; B.G. and S.R. wrote the manuscript; all authors read, commented on and approved the manuscript.

## Funding

## Competing interests

This work was supported by an Illumina research grant to AvD. Illumina was not involved in study design, collection, analysis, or interpretation of data, writing the manuscript, and the decision to submit the manuscript for publication. All other authors declare no competing interests.

## Additional information

¹Institute of Pathology, Heidelberg University, Medical Faculty, University Hospital Heidelberg, Heidelberg, Germany. ²Institute of Tissue Medicine and Pathology, University of Berne, Berne, Switzerland. ³Institute of Pathology, RKH Hospital Ludwigsburg, Ludwigsburg, Germany. ⁴Liver Cancer Center Heidelberg (LCCH), Heidelberg, Germany. ⁵Luxembourg Institute of Health, Department of Cancer Research, Strassen, Luxembourg. ⁶Clinical Cooperation Unit Neuropathology, German Cancer Research Center (DKFZ), Heidelberg, Germany. ⁷German Consortium for Translational Cancer Research (DKTK), Heidelberg, Germany. ⁸Department of Neuropathology, University Hospital Heidelberg, Heidelberg, Germany. ⁹Université Paris-Cité, Department of Pancreatology and Digestive Oncology, Beaujon Hospital (APHP) and INSERM U1149, Clichy, France. ¹⁰Université Paris-Cité, Department of Pathology, Beaujon Hospital (APHP) and INSERM U1149, Clichy, France. ¹¹Diagnostic and Interventional Radiology, Thoraxklinik at University Hospital Heidelberg, Heidelberg, Germany. ¹²Department of Radiology and Nuclear Medicine, Robert Bosch Hospital Stuttgart, Stuttgart, Germany. ¹³Department of Diagnostic and Interventional Radiology, Heidelberg University Hospital, Heidelberg, Germany. ¹⁴Department of Radiology and Nuclear Medicine, Luzerner

Kantonsspital, University Teaching and Research Hospital, University of Lucerne, Lucerne, Switzerland. [15]Tissue Bank of the National Center for Tumor Diseases (NCT), Heidelberg, Germany. [16]The MOE Key Laboratory of Biosystems Homeostasis & Protection, Zhejiang Provincial Key Laboratory for Cancer Molecular Cell Biology, and Innovation Center for Cell Signaling Network, Life Sciences Institute, Zhejiang University, Hangzhou, China. [17]Institute of Pathology and Neuropathology, Eberhard-Karls University, Tuebingen, Germany. [18]National Center for Tumor Diseases (NCT) Heidelberg, Department of Medical Oncology, University Hospital Heidelberg, Heidelberg, Germany. ✉e-mail: Benjamin.Goeppert@rkh-gesundheit.de; Stephanie.Roessler@med.uni-heidelberg.de

