## [Transparent Peer Review file · Nature Communications]

DNA methylation patterns facilitate tracing the origin of neuroendocrine neoplasms

Corresponding Author: Professor Stephanie Roessler

Version 0:

Reviewer comments:

Reviewer #1

(Remarks to the Author)

Comments on the manuscript entitled "DNA Methylation Patterns Facilitate Tracing the Origin of Neuroendocrine Neoplasms" by Goeppert et al.

The manuscript investigates the origin of neuroendocrine neoplasms (NEN) using DNA methylation profiling and machine learning algorithms on a dataset of 197 samples. The authors claim that DNA methylation profiles differ significantly across various anatomical locations, allowing for identification of tumor origins and clustering of primary tumor-metastasis pairs. They further use machine learning algorithms to predict the origin of NENs and claim achieving good prediction accuracy. The study lacks a clear explanation of the practical implications of these findings for clinical practice, possibly due to the lack of an external dataset for validation. In addition, the authors highlight a significant misclassification of hepatic NENs but do not provide a solution.

In conclusion, the study provides some insights and demonstrates the potential of DNA methylation analysis to predict the origin of NENs. However, it falls short in addressing the practical clinical applications and potential limitations of the methodology, which are essential for validating and implementing these findings in healthcare settings. The manuscript requires substantial revision before it can be considered for publication in a prestigious journal such as Nature Communications.

Other issues related to this manuscript are listed below:

1. The authors should briefly describe therapeutic strategies for hepatic NEN with different tumor origins and mention potential outcomes.
2. In lines 159-153, the authors state that "overall survival analysis showed that hepatic NEN without detectable primary tumor had a worse overall survival compared to patients with known NEN and liver metastasis." However, it is not clear whether the difference is caused by different organs of tumor origin or different treatments. Detailed information regarding how these patients are treated is needed.
3. The HCC(TCGA-PIHC) and CCA datasets are from published literature and were generated using the Illumina 450K platform, while the data in this study were generated using the Illumina 850K platform. The authors did not mention how they processed data generated from different technologies.
4. There is no proper explanation of how copy number alteration (CNA) analysis was performed. It is also interesting to know why hepatic NEN and NEN liver metastases show distinct CNA patterns, while pancreatic NEN and liver metastases of pancreatic NEN, as well as NEN of the ileum and liver metastases of ileal NEN, illustrate similar CNA profiles.
5. When characterizing the organ origin of hepatic NEN using methylation profiles, the authors introduce a term "latent methylation components (LMC)," which is not a commonly used epigenetic term. However, they failed to provide further information regarding how LMC was determined. The authors used deconvolution analysis to identify 10 LMCs associated with specific NEN subgroups. Machine learning algorithms trained on these LMCs achieved high prediction accuracies, correctly classifying the organ of origin for NENs, including hepatic NENs. The authors claimed that DNA methylation analysis is a feasible method for tracing the origin of NENs. However, it is worth pointing out that the authors used a small

cohort of patients and failed to use an external cohort to verify this conclusion.

6. All data should be stored in a public domain for evaluation, rather than waiting for reviewers/readers' requests.

Reviewer #2

(Remarks to the Author)

Reviewer #3

(Remarks to the Author)

In this paper the authors report their studies to characterize the DNA methylation pattern of 185 neuroendocrine neoplasms (NENs) with a primary focus on liver tumors that included 22 metastatic foci and 22 without a known extrahepatic primary tumor. They identify the DNA methylation patterns of liver metastases of NENs from the most common primary sites (pancreas, ileum, appendix, colorectum and lung). They propose that this approach can be used to identify tumor origin.

The paper is interesting but there are several issues that should be addressed to improve the clinical value of this work.

1. The mix of NETs and NECs is problematic. These should be separated for better clarification of the results. Merkel cell carcinomas should be included in the NEC category.

2. The initial workup of the cases is insufficient and the value of the study is overblown by the lack of appropriate information about the tumors. It is not sufficient to assess a NET only by expression of chromogranin and synaptophysin. It is well known that NETs in particular express transcription factors that are reflective of their site of origin: TTF1 (using the SPT24 clone) for lung, ARX/PDX for pancreas, CDX2 for gut, SATB2 for distal ileum, appendix and large bowel, etc. Other helpful biomarkers include hormones and enzymes that are expressed at specific sites (e.g. CEA for thyroid NETs i.e. medullary thyroid carcinoma). The possibility of paraganglioma must also be assessed by expression of GATA3 and lack of keratin staining as well as expression of tyrosine hydroxylase. This should be done to validate all the cases. The most important scenario here is for the supposed "primary hepatic" tumors as these are likely to represent paragangliomas of the porta hepatis.

3. There is an issue with "2 out of 9 NEN of the ascending colon clustered together with ileal NEN, whereas 1 NEN of the ascending colon showed appendiceal similarity". This problem should be addressed by assessing these tumors for SATB2 and serotonin; these may all be EC cell tumors of the small bowel or appendix with deposits involving the ascending colon. Staining for SATB2 will support the origin in the very distal ileum or appendix.

4. The authors refer to "Poorly differentiated NET G3 and NEC". This is incorrect. NET G3 is by definition well differentiated whereas any poorly differentiated tumor is NEC. These should be verified by genetic profiling and expression of SSTRs.

5. The concept of "foregut" tumors is rather antiquated. It is now well recognized that this is a reflection of cell type with the commonest cell types in the various locations representing the "foregut" or "midgut" or "hindgut" tumors. For example the serotonin-producing EC cell tumors are the most common ileal tumors by far and represent the classic "midgut" tumor.

6. Please compare the accuracy of "17 of 22 (77.3%) NEN liver metastases" classified by this technique with that of a full immunohistochemical classification using transcription factors and hormones etc. This must be done properly before claiming that "identifying the primary tumor through conventional histopathology and immunohistochemistry is often not possible using conventional histopathological and immunohistochemical tools" as it appears that such tools were NOT used for the cases in this study.

Reviewer #4

(Remarks to the Author)

Description of the work

The authors performed methylation analysis on a large set of NEN samples (N=197) arising from different anatomic sites. These included 153 primary NENs, 22 liver metastases from different non-hepatic NENs and 22 samples classified as hepatic NENs due to the lack of any clinical evidence of another possible primary site.

By applying machine learning algorithms, the authors analyzed the performance of methylome-based classification of NEN from different sites, and its capacity to predict a possible extrahepatic origin of the 22 hepatic NENs, assigning a precise anatomic site based on the specific methylation profile of each sample.

Reviewer's comment

The aim of the work is very interesting as it covers a controversial topic in the classification of NENs. The manuscript is generally well presented but needs clarifying several points.

1. Introduction, page 4 last paragraph, fourth line – row 107 ("Thus, we aimed to identify the morphomolecular..."). Based on the results presented, this is almost exclusively a molecular (i.e. epigenomic) analysis. Pathological data were mainly used to present the cohort and then no longer used. Moreover, the "involvement in NEN development" does not appear to be a real aim of the manuscript; rather the main focus is on the performance of methylome analysis for the classification of NENs. I suggest rewording this sentence. Moreover, the citation of a figure (which is not even the first figure of the manuscript but figure 1B) in the Introduction is inappropriate.
2. The last sentence of the introduction may be removed, as it is already delving into the results and conclusions that are at least puzzling in this sentence... stating that methylation suggest potential tumor origin and this "potential" may lead to subsequent novel classification applying machine learning algorithms (!?). There is not data in this paper related to classification and artificial intelligence.
3. Results, Clinicopathological characteristics of the study cohort (page 5). Despite the efforts of the authors and the diagram of figure 1B, it is still difficult to spot cases with matched primary/metastasis or cases with multiple samples. I recommend changing figure 1b to better show matched samples.
4. In the second paragraph of the same subsection the information about Ki-67 staining and tumor grade is confused: tumor grade is used but never introduced, and Ki-67 is generically described as "low to intermediate and high proliferation rate" instead of introducing and using the WHO-defined cut-offs. As for MiNEN, the description should better clarify here, and not later in the manuscript, that the neuroendocrine component only was analyzed. Finally, the difference in survival reflects the fact that 45.5% of Hepatic NEN are neuroendocrine carcinomas, while only 86.7% of liver metastases are NETs. I recommend rewriting this paragraph to increase its clarity and removing the survival analysis.
5. Table 3 is missing the "NET" definition in the "G" panel
6. Please briefly introduce tSNE analysis at the beginning of the molecular analysis results (page 6).
7. Results, "DNA methylation-based analysis shows distinct grouping for NEN of most organ sites" (page 6). The inclusion of HCC and CCA in the analysis is puzzling, as they were only briefly cited in the introduction as the exocrine counterpart to hepatic MiNEN. Moreover, they are retained in the tSNE analysis throughout all figures. Finally, these samples come from previous publications. I recommend clarifying the use of these samples within the aim of the paper and explain the strategy used to normalize the data produced in this publication with those from previous ones. Additionally, I would suggest keeping HCC and CCA data only in figure 3, since the rest of the comparison only involves NET/NEC cases.
8. Figure 2B: I recommend adding the primary tumor site and grouping patients accordingly.
9. While the authors do not consider the neoplastic cellularity (cancer cell fraction) of the samples, I recommend including this information as it may provide a clue to interpret some discrepancies in the data. For instance, those between the tSNE analysis and the hierarchical clustering of paired primary/metastasis cases.
10. Results, "Epigenetic profiling reveals distinct clusters of extrahepatic NEN, hepatic NEN, HCC and CCA" (page 7). The description of poorly and moderately differentiated NETs is not in line with what is reported in the current WHO classification of neuroendocrine tumors. Please have the manuscript reviewed by a pathologist with expertise in surgical pathology of neuroendocrine neoplasms. If he/she is among the authors, involve her/him actively in the revision of the work (see also points above). This could also provide more in-depth insights throughout the manuscript and more focused discussions and certainly raise the level of the manuscript.
11. Results, "NEN of different organ sites, hepatic NEN and NEN metastases exhibit distinct genetic profiles" (page 8) This subsection interrupts the methylation analysis while adding little incremental information. In lack of a better integration with the rest of the data, I recommend removing this analysis or moving most of it to supplementary information, also changing its position within the manuscript to avoid a sudden change of the focus from methylation to copy number analysis.
12. Results, "Most hepatic NEN are predicted to be of non-hepatic origin and show a foregut methylation pattern" (page 8) Again, the authors quickly dismiss tumor purity (that is, neoplastic cellularity). I recommend including it in the analysis (see point 9).
13. Figure 5D: it would be nice to include also a modified version of this figure where hepatic NEN samples are placed near to the primary NEN they were predicted to originate from.
14. Table 4: Please add a legend to the abbreviations. The authors should explain in the results how the prediction is made when the prediction scores are similar for different sites (e.g. patient 14). Is there a cut-off or any other tool to judge the confidence of single predictions?

Version 1:

Reviewer comments:

Reviewer #1

(Remarks to the Author)

The authors have addressed all of my questions and concerns. I have no additional comments.

(Remarks on code availability)

Reviewer #2

(Remarks to the Author)

(Remarks on code availability)

The code includes a README file with sufficient instructions for installing and running the application, and I was able to install and execute the code successfully.

Reviewer #4

(Remarks to the Author)

The authors responded adequately to all the criticisms and appropriately amended the manuscript.

(Remarks on code availability)

The authors responded adequately to all the criticisms and appropriately amended the manuscript.

Reviewer's comments and point-by-point reply:

Reviewer #1 (Remarks to the Author): expert in DNA methylation and classifiers, machine learning

Comments on the manuscript entitled "DNA Methylation Patterns Facilitate Tracing the Origin of Neuroendocrine Neoplasms" by Goeppert et al.

The manuscript investigates the origin of neuroendocrine neoplasms (NEN) using DNA methylation profiling and machine learning algorithms on a dataset of 197 samples. The authors claim that DNA methylation profiles differ significantly across various anatomical locations, allowing for identification of tumor origins and clustering of primary tumor-metastasis pairs. They further use machine learning algorithms to predict the origin of NENs and claim achieving good prediction accuracy. The study lacks a clear explanation of the practical implications of these findings for clinical practice, possibly due to the lack of an external dataset for validation. In addition, the authors highlight a significant misclassification of hepatic NENs but do not provide a solution.

In conclusion, the study provides some insights and demonstrates the potential of DNA methylation analysis to predict the origin of NENs. However, it falls short in addressing the practical clinical applications and potential limitations of the methodology, which are essential for validating and implementing these findings in healthcare settings. The manuscript requires substantial revision before it can be considered for publication in a prestigious journal such as Nature Communications.

We thank the reviewer for the suggestions to perform additional analyses and thereby corroborating our results and strengthening our hypotheses. In the revised manuscript, we have employed an external validation cohort, added detailed information regarding the methods and outlined the clinical relevance of our study.

We performed various additional analyses including global DNA methylation analyses of a completely independent validation cohort of hepatic NEN without any detectable primary. This independent hepatic NEN cohort originated from Beaujon Hospital, Paris, France. Thereby, we could show that this cohort shows generally comparable results to our cohort of hepatic NEN. In detail, the hepatic NEN of the Beaujon cohort showed that hepatic NEN did not cluster as one group but showed various DNA methylation patterns partly clustering with established extrahepatic NEN profiles of our presented NEN classifier. In conclusion, the results of the validation cohort matched the results of the Heidelberg discovery cohort. In addition, we added a comprehensive immunohistochemical characterization of the clinically well-characterized Heidelberg cohort, confirming the clinical data.

In conclusion, the presented study shows i) that methylome analysis is a powerful tool to detect the primary tumor in neuroendocrine neoplasms, ii) that hepatic NEN most likely do not arise in the liver, but rather represent metastases of primarily clinically undetectable extrahepatic tumors and iii) that distinct clinical differences exist between NET and NEC in the liver (please also refer to Reviewer #3, point 1). These findings have not only implications to diagnostic algorithms but also to clinical management of patients with neuroendocrine neoplasms. Adding methylome analyses to established immunohistochemical markers might therefore help identifying the potential primary and adjust treatment strategies accordingly (please refer to point 1 of the point-by-point-reply).

Other issues related to this manuscript are listed below:

1. The authors should briefly describe therapeutic strategies for hepatic NEN with different tumor origins and mention potential outcomes.

We have now included a paragraph in the introduction describing the therapeutic strategies for hepatic NEN with different tumor origins (page 4). Since there is no specific evidence for optimal management of primary hepatic NEN, treatment is performed in analogy to NEN with gastrointestinal primaries. This includes evaluation of resectability of hepatic tumor manifestation as well as liver transplantation in selected cases. For systemic treatment, platinum and etoposide-based chemotherapy is applied in high-grade hepatic neuroendocrine carcinomas, whereas mainly somatostatin analogues, peptide receptor radionuclide therapy and targeted agents such as everolimus are applied in well-differentiated hepatic NET. All patients included in this study were discussed in a multidisciplinary tumor board (Material and Methods, page 15). In addition, we emphasized the importance of identifying the primary tumor in management of hepatic manifestations of NEN (page 4).

2. In lines 159-153, the authors state that “overall survival analysis showed that hepatic NEN without detectable primary tumor had a worse overall survival compared to patients with known NEN and liver metastasis.” However, it is not clear whether the difference is caused by different organs of tumor origin or different treatments. Detailed information regarding how these patients are treated is needed.

Regarding locoregional treatment, 6 (27.3%) patients with hepatic NEN of the discovery cohort (Heidelberg cohort) received partial liver resection vs 12 (54.5%) patients with metastatic NEN to the liver. Those patients also received a resection of their detectable primaries. Liver transplantation was performed in one patient with hepatic NEN and two patients with extrahepatic NEN. Selective internal radiotherapy (SIRT) was only performed in liver metastases of non-hepatic NEN (n=3). Regarding systemic treatment, chemotherapy was applied to both a high proportion of the primary hepatic NEN (n=10) and the group of metastatic NEN to the liver (n=12). Targeted therapies, employing everolimus, antiangiogenic drugs or immune checkpoint inhibitors was applied to a small proportion of both groups (n=3 vs n=5). However, less primary hepatic NEN received somatostatin analogues (n=3 vs n=11) or peptide receptor radionuclide therapy (n=1 vs n=6). In the revised manuscript, the therapeutic regimens of patients with hepatic NEN or NEN liver metastases were included in Table 1.

In summary, the lower proportion of patients receiving surgery or somatostatin receptor-based treatments in the group of primary hepatic NEN might reflect a more aggressive tumor biology including a higher proportion of poorly differentiated neuroendocrine carcinomas in this group. In the new Supplemental Figure S1B, we could show that although the differences in grading were not statistically significant (Table 1), hepatic NEC exhibit significantly worse overall survival compared to hepatic NET. Therefore, the survival differences between patients with hepatic NEN and liver metastasis seem to be mainly determined by tumor subtyping (Supplemental Figure S1). A discussion of these findings was added on pages 12 and 13.

3. The HCC (TCGA-LIHC) and CCA datasets are from published literature and were generated using the Illumina 450K platform, while the data in this study were generated using the Illumina 850K platform. The authors did not mention how they processed data generated from different technologies.

Illumina EPIC and 450K methylation array samples were merged into a single dataset using their common probes according to the manufacturer's annotations using the minfi package. Each sample then underwent individual normalization, which included a background correction and a dye-bias correction (scaling normalization control probe intensities). Following this, a batch correction was applied to the log₂-transformed intensity values using univariable linear models with the limma package. This step adjusted for variations of tissue type (FFPE or frozen) and array type (450K or EPIC), with methylated and unmethylated signals corrected separately. Finally, beta-values were calculated from the re-transformed intensities, incorporating an offset of 100. The CpG probes are filtered according to the filter criteria described in Patel et al. ¹.

We included this detailed information in the Materials and Methods section, see page 17.

4. There is no proper explanation of how copy number alteration (CNA) analysis was performed. It is also interesting to know why hepatic NEN and NEN liver metastases show distinct CNA patterns, while pancreatic NEN and liver metastases of pancreatic NEN, as well as NEN of the ileum and liver metastases of ileal NEN, illustrate similar CNA profiles.

We have now included in the revised manuscript a detailed description of how copy number alteration (CNA) analysis was performed in the Materials and Methods section (page 18). In previous studies and in this study, we used the R package *conumee* (<http://bioconductor.org/packages/conumee/>) for CNA analysis ^{2,3,4}. Every interrogated CpG of the Illumina DNA methylation microarrays is represented by two probes on the array. One of these probes detects the methylated and the other one the unmethylated CpG. For the analysis of DNA methylation, the ratio of the intensity signal of the methylated (M) and the sum of the methylated (M) and unmethylated probe intensities (U) are calculated as $M/(M+U)$ =beta-value. In contrast, for the calculation of CNA, the methylated and unmethylated signal intensities are added together and a ratio is formed against healthy reference samples that have a flat genome. This copy-number ratio is then plotted in a graph according to chromosomal location. The *conumee* package has been downloaded almost 20,000 times, is one of the most widely used tools for inferring CNA from DNA methylation arrays and has been applied in numerous large-scale cancer research projects ^{5,6,7}.

Large CNA have been shown to occur early during carcinogenesis and patterns of CNA are tumor-type specific ⁸. Although intra-tumor heterogeneity has been observed, most CNA are stable over time and the majority of CNA are preserved during cancer progression ^{9,10,11}. However, during therapy subclones may be selected for therapy resistance. Consistently, the pancreatic NEN and liver metastases of pancreatic NEN, as well as NEN of the ileum and liver metastases of ileal NEN, illustrate similar CNA profiles.

The analysis of NEN liver metastases did not show strong similarity with any of the reference groups consistent with the heterogeneous origin of the metastases (Supplemental Figure S7). Thus, the NEN liver metastases show a mixture of CNA profiles stemming from the different origins. Similarly, CNA of the hepatic NEN did not exhibit strong similarity to any of the reference NEN groups suggesting that the hepatic NEN are a heterogeneous group of NEN as well. As the hepatic NEN and NEN liver metastasis differ in their clinical characteristics, i.e. NET and NEC, thus and the origin and the CNA profiles differ between NEN liver metastases and hepatic NEN.

5. When characterizing the organ origin of hepatic NEN using methylation profiles, the authors introduce a term “latent methylation components (LMC),” which is not a commonly used epigenetic term. However, they failed to provide further information regarding how LMC was determined. The authors used deconvolution analysis to identify 10 LMCs associated with specific NEN subgroups. Machine learning algorithms trained on these LMCs achieved high prediction accuracies, correctly classifying the organ of origin for NENs, including hepatic NENs. The authors claimed that DNA methylation analysis is a feasible method for tracing the origin of NENs. However, it is worth pointing out that the authors used a small cohort of patients and failed to use an external cohort to verify this conclusion.

Hepatic NEN are a rare entity and only few specialized centers exist. Louis de Mestier and Jérôme Cros are internationally renowned experts in neuroendocrine tumors at the ENETS Centre of Excellence in Paris France. They provided us with an independent cohort of hepatic NEN which they have previously comprehensively analyzed¹². We performed DNA methylation analysis of this independent cohort including 15 hepatic NEN. Similar to the hepatic NEN of our cohort, the hepatic NEN of the external validation cohort from Paris did not form a distinct cluster in t-SNE plots and showed highly overlapping patterns with our hepatic NEN. In addition, our NEN classifier predicted non-hepatic origin including appendix (N=2), gastric/duodenal (N=9), ileum (N=2) and lung (pulmonary carcinoid, N=2) of the external validation cohort from Paris. This result mirrored our prediction results of the NET cases in our discovery cohort from Heidelberg. See Figure 6 and Tables S4 and S7 for details.

We included Louis de Mestier and Jérôme Cros as co-authors in the revised manuscript for their contributions.

6. All data should be stored in a public domain for evaluation, rather than waiting for reviewers/readers' requests.

We agree that data availability is a high-ranked priority. We therefore included a reviewer token in the revised manuscript for GSE253176 under the section Data availability. In addition, a reviewer token is also provided in the same section for the new validation cohort which has the GEO series number GSE298677.

As soon as the manuscript has been accepted, we will make all DNA methylation data publicly available as we have done for all our previous publications, GSE201241 and GSE156299^{3, 4}.

Reviewer #2 (Remarks to the Author)

Reviewer #3 (Remarks to the Author)

In this paper the authors report their studies to characterize the DNA methylation pattern of 185 neuroendocrine neoplasms (NENs) with a primary focus on liver tumors that included 22 metastatic foci and 22 without a known extrahepatic primary tumor. They identify the DNA methylation patterns of liver metastases of NENs from the most common primary sites (pancreas, ileum, appendix, colorectum and lung). They propose that this approach can be used to identify tumor origin.

We thank the reviewer for the opportunity to conduct additional analyses, which have confirmed our findings and further strengthened our hypotheses. In the revised manuscript, we incorporated an external validation cohort from Paris, provided a detailed immunohistochemical characterization of more than 900 immunohistochemical stainings, and included information on the methods and the clinical significance of our study.

The paper is interesting but there are several issues that should be addressed to improve the clinical value of this work.

1. The mix of NETs and NECs is problematic. These should be separated for better clarification of the results. Merkel cell carcinomas should be included in the NEC category.

We completely agree that the NEN subgroups NET G3 and NEC need to be distinguished as clearly as possible. Therefore, we performed more than 900 supplementary immunohistochemical analyses (Table S1, S2, S3, S4, Supplemental Figure S2E-G). In particular, immunohistochemistry of p53 and RB1 have been shown to be able to separate these histopathologically intermingling groups of NET G3 and NEC, and therefore can be used to maximize the discriminatory power between NET G3 and NEC. Following this approach, we were able to clearly distinguish NET G3 and NEC. On this basis, the DNA methylation results are applicable to these subgroups and provide valuable additional information not previously described in these NEN subgroups.

We have added the results of the additional immunohistochemical analyses in a new supplemental table, which is now included in the Supplemental Material (Table S2). In the revised manuscript, we added the overall survival analysis of hepatic NET and hepatic NEC which showed that hepatic NEC exhibit worse outcome (Supplemental Figure S1). Moreover, DNA methylation and copy number alteration patterns were re-analyzed using the subgrouping of NET G3 and NEC (Supplemental Figure S5 and S6).

2. The initial workup of the cases is insufficient and the value of the study is overblown by the lack of appropriate information about the tumors. It is not sufficient to assess a NET only by expression of chromogranin and synaptophysin. It is well known that NETs in particular express transcription factors that are reflective of their site of origin: TTF1 (using the SPT24 clone) for lung, ARX/PDX for pancreas, CDX2 for gut, SATB2 for distal ileum, appendix and large bowel, etc. Other helpful biomarkers include hormones and enzymes that are expressed at specific sites (e.g. CEA for thyroid NETs i.e. medullary thyroid carcinoma). The possibility of paraganglioma must also be assessed by expression of GATA3 and lack of keratin staining as well as expression of tyrosine hydroxylase. This should be done to validate all the cases. The most important scenario here is for the supposed “primary hepatic” tumors as these are likely to represent paragangliomas of the porta hepatis.

We thank the reviewer for this valuable suggestion and fully agree that there were some immunohistochemical gaps of our clinically well-characterized NEN cohort. Regarding

different antibody clones against TTF1, we used the clone SP141 which was shown to have highest sensitivity and overall similar sensitivity compared to clone ST24^{13, 14}. See Table S8 for a list of the antibodies used in this study. To confirm the clinical data and to strengthen our study, we have performed more than 900 additional immunohistochemical analyses for all available cases (including TTF1, ARX, PDX, CDX2, SATB2, Calcitonin, Thyroglobulin, GATA3 and CK). Thereby, the subgrouping of our cohort was confirmed. In addition, the presented methylome data is in line with the additional immunohistochemical results. In conclusion, we believe that our study now provides sufficient data for a robust classification of NEN. The additional analyses are now included in the revised manuscript as Tables S1-4, S8 and Supplemental Figure S2.

3. There is an issue with “2 out of 9 NEN of the ascending colon clustered together with ileal NEN, whereas 1 NEN of the ascending colon showed appendiceal similarity”. This problem should be addressed by assessing these tumors for SATB2 and serotonin; these may all be EC cell tumors of the small bowel or appendix with deposits involving the ascending colon. Staining for SATB2 will support the origin in the very distal ileum or appendix.

We fully agree that further immunohistochemistry could provide greater separation. Therefore, we performed additional SATB2 immunohistochemistry for representative NEN of the colon and the 2 NEN which clustered together with the ileal or appendiceal NEN (Table S2). All NEN of the colon exhibited nuclear SATB2 staining as shown in representative immunohistochemical microphotographs (Supplemental Figure S2E-G).

4. The authors refer to “Poorly differentiated NET G3 and NEC”. This is incorrect. NET G3 is by definition well differentiated whereas any poorly differentiated tumor is NEC. These should be verified by genetic profiling and expression of SSTRs.

We have corrected this misclassification in the appropriate places in the revised manuscript. As stated in point 1, we have now separated the groups of NET G3 and NEC as clearly as possible by additional immunohistochemical analyses. In addition, all NET G3 and NEC cases were reviewed by two diagnostic pathologists with a special expertise in NEN (BG and AP).

We performed additional genetic analyses of NEC and NET G3 (copy number alterations, CNA). The CNA profiles showed differences between the two groups mainly including chromosome 3q (Supplemental Figure S6).

Of the 11 hepatic NEC in our cohort, 9 (82%) were SSTR2A positive by immunohistochemical staining and 2 were negative. Similarly, of the 3 hepatic NET G3, 2 cases showed positive SSTR2A staining and for one sample immunohistochemical analysis could not be performed due to lack of material (Table S3). This result is consistent with literature concerning SSTR2 expression in NET G3 and NEC¹⁵.

5. The concept of “foregut” tumors is rather antiquated. It is now well recognized that this is a reflection of cell type with the commonest cells types in the various locations representing the “foregut” or “midgut” or “hindgut” tumors. For example the serotonin-producing EC cell tumors are the most common ileal tumors by far and represent the classic “midgut” tumor.

We fully agree that the separation of NEN into these three types (foregut, midgut and hindgut tumors), which has been frequently used by various authors to date, is rather imprecise and

that a tumor classification according to the organ of origin and neuroendocrine cell-of-origin is more precise. Transcription factor expression correlates with both, organ of origin and neuroendocrine cell-of-origin, and additional immunohistochemical analyses have been performed. Thus, we addressed this important point in the introduction of the revised manuscript (page 3). As the cell-of origin concept is still debated, the main focus in the text is put onto organ-of-origin, which is still the main concept used for therapy indication and clinical trial design.

We achieved this clinically, diagnostically and putative therapeutically relevant goal by means of matched-pair analyses of known tumors and the corresponding liver metastases. In addition, we were able to show that the cases previously clinically classified as hepatic NEN not only did not show any new cluster formation, but could also be re-assigned in part to other known clusters of extrahepatic primary NEN. Future work should investigate the interesting aspect of subtyping based on cell-of-origin in NEN, and DNA methylation profiles could be a powerful tool in this attempt. The prerequisite for this approach would be to know the cell-of-origin DNA methylation profiles of the different non-neoplastic cell types (e.g. ECL for gastric NEN) in advance and then test unknown NEN of various localizations and subtypes; unfortunately, such data is not available yet.

6. Please compare the accuracy of “17 of 22 (77.3%) NEN liver metastases” classified by this technique with that of a full immunohistochemical classification using transcription factors and hormones etc. This must be done properly before claiming that “identifying the primary tumor through conventional histopathology and immunohistochemistry is often not possible using conventional histopathological and immunohistochemical tools” as it appears that such tools were NOT used for the cases in this study.

We absolutely agree and have therefore subjected the entire NEN cohort to extensive supplementary immunohistochemical analysis. In detail, we performed more than 900 immunohistochemical analyses. These fully confirmed the in our manuscript presented clinical characterization of the cohort. In addition, differential diagnoses (paraganglioma, medullary thyroid carcinoma, etc.) and NEN subtyping questions (e.g. NET G3 vs NEC) could be clarified. Taken together, we used an additional panel of immunohistochemical markers (TTF1, ARX, PDX, CDX2, SATB2, GATA3, Thyreoglobulin, Calcitonin, CK, RB1 and p53) to comprehensively characterize our cohort, as suggested by you and reviewer #1 and #4 (Table S1-S4, Supplemental Figure S2E-G).

In conclusion, the clinical and pathological characterization of our study cohort has now significantly improved.

Eva-Marie Bohn supported the additional analyses and was therefore included as co-author.

Reviewer #4 (Remarks to the Author): expert in NEN classification, Hepatic NEN

Description of the work

The authors performed methylation analysis on a large set of NEN samples (N=197) arising from different anatomic sites. These included 153 primary NENs, 22 liver metastases from different non-hepatic NENs and 22 samples classified as hepatic NENs due to the lack of any clinical evidence of another possible primary site.

By applying machine learning algorithms, the authors analyzed the performance of methylome-based classification of NEN from different sites, and its capacity to predict a possible extrahepatic origin of the 22 hepatic NENs, assigning a precise anatomic site based on the specific methylation profile of each sample.

Reviewer's comment

The aim of the work is very interesting as it covers a controversial topic in the classification of NENs. The manuscript is generally well presented but needs clarifying several points.

We thank the reviewer for this assessment and critical evaluation. In the revised manuscript, we addressed all points raised by the reviewer which significantly helped to improve the manuscript.

1. Introduction, page 4 last paragraph, fourth line – row 107 ("Thus, we aimed to identify the morphomolecular...").

Based on the results presented, this is almost exclusively a molecular (i.e. epigenomic) analysis. Pathological data were mainly used to present the cohort and then no longer used. Moreover, the "involvement in NEN development" does not appear to be a real aim of the manuscript; rather the main focus is on the performance of methylome analysis for the classification of NENs. I suggest rewording this sentence. Moreover, the citation of a figure (which is not even the first figure of the manuscript but figure 1B) in the Introduction is inappropriate.

We agree that our aim is primarily to use DNA methylation profiles to better trace the origin of NEN. Accordingly, we have made the changes in the introduction as suggested and changed "morphomolecular" to "molecular". In addition, we removed Figure 1B from the introduction and reference this figure in the results section only as remarked.

2. The last sentence of the introduction may be removed, as it is already delving into the results and conclusions that are at least puzzling in this sentence... stating that methylation suggest potential tumor origin and this "potential" may lead to subsequent novel classification applying machine learning algorithms (!?). There is not data in this paper related to classification and artificial intelligence.

As this sentence could be misinterpreted, we changed it accordingly. We performed deconvolution analysis (MeDeCom) which was the basis for the development of a Random Forest classifier (Figure 5). The use of Random Forest is regarded a machine learning algorithm. For improved clarity, we changed the last sentence of the introduction to "Furthermore, deconvolution analysis identified latent methylation components (LMC) and potential tumor origin sites, thereby providing a novel classification by DNA methylation profiling. In particular, DNA methylation analysis could identify extrahepatic origin of a

substantial proportion of hepatic NEN that were clinicopathologically classified as primary hepatic NEN.”. In addition, we added further detail to the Results “Next, LMC were used to train Random Forest machine learning algorithms for classification of NEN subgroups (see Materials and Methods for details).” (page 9).

3. Results, Clinicopathological characteristics of the study cohort (page 5). Despite the efforts of the authors and the diagram of figure 1B, it is still difficult to spot cases with matched primary/metastasis or cases with multiple samples. I recommend changing figure 1b to better show matched samples.

As we now included an independent validation cohort from Beaujon hospital, Paris, France, we changed Figure 1B accordingly. For a better understanding of the matched pair samples and for reasons of space, a diagram is shown in Supplemental Figure S3A of the revised manuscript.

4. In the second paragraph of the same subsection the information about Ki-67 staining and tumor grade is confused: tumor grade is used but never introduced, and Ki-67 is generically described as "low to intermediate and high proliferation rate" instead of introducing and using the WHO-defined cut-offs. As for MiNEN, the description should better clarify here, and not later in the manuscript, that the neuroendocrine component only was analyzed. Finally, the difference in survival reflects the fact that 45.5% of Hepatic NEN are neuroendocrine carcinomas, while only 86.7% of liver metastases are NETs. I recommend rewriting this paragraph to increase its clarity and removing the survival analysis.

The WHO-defined cut-offs for Ki67-based proliferation rates of neuroendocrine neoplasms were applied in this study. The cut-offs have now been included in the respective results section of the manuscript (page 5). Furthermore, the WHO-defined Ki67 cut-offs and respective grouping are listed in Table 1, 2 and 3.

As suggested, we included the information, that only the neuroendocrine component of MiNEN cases was analyzed, in the results section on pages 6 and 7.

To elucidate the differences in patient survival in the hepatic NEN group, we performed subgroup analysis (NET vs NEC) of overall patient survival. In fact, the differences in survival between patients with hepatic NEN and liver metastasis of NEN can be explained by significant outcome differences between NET and NEC. A similar question was also raised by reviewer #1, see point 2 of reviewer #1 above. In the new Supplemental Figure S1B, we could show that patients with hepatic NEC exhibit significantly worse overall survival compared to hepatic NET, although the differences in grading were not statistically significant between hepatic NEN and NEN with liver metastasis (Table 1). Therefore, the survival differences between patients with hepatic NEN and NEN liver metastasis seem to be mainly determined by tumor subtyping (NET vs NEC). We rewrote this paragraph and included these additional findings (page 6; Supplemental Figure S1).

5. Table 3 is missing the "NET" definition in the "G" panel

Thank you for pointing this out. We added the NET definition to Table 3.

6. Please briefly introduce tSNE analysis at the beginning of the molecular analysis results (page 6).

For a better understanding, we have now added a brief introduction to t-SNE analysis in the Results section (page 6). "For visualization of high-dimensional data, t-SNE plots assign each sample a location in a two-dimensional map in such a way that similar samples are modeled by nearby points and dissimilar samples are modeled by distant points."

7. Results, "DNA methylation-based analysis shows distinct grouping for NEN of most organ sites" (page 6). The inclusion of HCC and CCA in the analysis is puzzling, as they were only briefly cited in the introduction as the exocrine counterpart to hepatic MiNEN. Moreover, they are retained in the tSNE analysis throughout all figures. Finally, these samples come from previous publications. I recommend clarifying the use of these samples within the aim of the paper and explain the strategy used to normalize the data produced in this publication with those from previous ones. Additionally, I would suggest keeping HCC and CCA data only in figure 3, since the rest of the comparison only involves NET/NEC cases.

As the current concept implies that hepatic NEN, also named primary hepatic NEN may originate from the liver, and in addition MiNEN have been described in the liver in combination with HCC and CCA, we included the DNA methylation profiles of these two most prevalent primary liver tumors for comparative purposes. Thus, we aimed to decipher the origin of NEN within the liver. This was pointed out in the revised manuscript on page 8. As t-SNE plots apply high-dimensional data, high similarity due to the same origin results in close proximity within t-SNE plot. Thus, we included HCC and CCA samples to interrogate the potential common origin of hepatic NEN and HCC or CCA. Therefore, HCC and CCA samples were included in the t-SNE plots for visualization. However, we excluded HCC and CCA samples from training the Random Forest algorithm to predict the NEN origin.

We generated the CCA dataset in a previous study using the same DNA methylation platform as in the present study (Illumina MethylationEPIC)³. The HCC data were retrieved from TCGA-LIHC which were performed using Illumina Human Methylation 450 (450K arrays). In the revised manuscript, we included detailed information regarding the data normalization. Illumina EPIC and 450K methylation array samples were merged into a single dataset using their common probes according the manufactures annotations using the minfi package. Each sample then underwent individual normalization, which included a background correction and a dye-bias correction (scaling normalization control probe intensities). Following this, a batch correction was applied to the log₂-transformed intensity values using univariable linear models with the limma package. This step adjusted for variations of tissue type (FFPE or frozen) and array type (450K or EPIC), with methylated and unmethylated signals corrected separately. Finally, beta-values were calculated from the re-transformed intensities, incorporating an offset of 100. The CpG probes are filtered according the filter criteria described in Patel et al.¹.

However, in order to avoid confusion, we have now removed the HCC and CCA data sets from Figure 4 and moved this data to the new Supplemental Figure S7.

8. Figure 2B: I recommend adding the primary tumor site and grouping patients accordingly.

We agree completely and have now improved the quality of Figure 2B by enhancing the color coding. In addition, we have now included a detailed depiction of the matched paired analysis for better understanding in Supplemental Figure S3.

9. While the authors do not consider the neoplastic cellularity (cancer cell fraction) of the samples, I recommend including this information as it may provide a clue to interpret some discrepancies in the data. For instance, those between the tSNE analysis and the hierarchical clustering of paired primary/metastasis cases.

We added in the Materials and Methods section under “Genomic DNA isolation” the approach taken to ensure similar tumor cell fractions. To ensure high tumor cell fraction, all NEN samples were pathologically evaluated for regions with highest tumor content. Thereby, no differences between primary tumor and metastasis were observed. The fraction of immune cell infiltration and tumor stroma were highly similar between primary tumor and metastasis. For DNA methylation analyses, tumor areas with high tumor cell content and low stromal and immune cell infiltration were selected. In addition, for all cases tumor cell content was >50%. Furthermore, tumor cell purity was confirmed by Leukocytes Unmethylation for Purity (LUMP) analysis¹⁶ based on the DNA methylation profiles (Supplemental Figure S13). LUMP is based on the detection of immune cell specific DNA methylation sites and widely used to measure tumor cell purity. Supplemental Figure S13 shows that LUMP varies only to a small degree between NEN subgroups. However, we observed that LUMP correlates with LMC2 and therefore, for prediction of tumor origin, methylation sites correlating with LMC2 were excluded to ensure the detection of tumor cell signatures only. As suggested, we have included the LUMP of each sample in Table S5.

In addition, comparative SNP analysis was conducted revealing high similarity for all paired samples which confirmed sample identity and indicated molecular changes associated with tumor progression (Supplemental Figure S4).

10. Results, "Epigenetic profiling reveals distinct clusters of extrahepatic NEN, hepatic NEN, HCC and CCA" (page 7). The description of poorly and moderately differentiated NETs is not in line with what is reported in the current WHO classification of neuroendocrine tumors. Please have the manuscript reviewed by a pathologist with expertise in surgical pathology of neuroendocrine neoplasms. If he/she is among the authors, involve her/him actively in the revision of the work (see also points above). This could also provide more in-depth insights throughout the manuscript and more focused discussions and certainly raise the level of the manuscript.

We thank the reviewer for this remark and have corrected the description regarding poorly and moderately differentiated NET according to the current WHO classification of neuroendocrine tumors.

In addition, we completely agree that the NEN subgroups NET G3 and NEC need to be distinguished as clearly as possible. Immunohistochemistry of p53 and RB1 has been shown to be reliable in separating these histopathologically intermingling groups of NET G3 and NEC, and therefore can be used to maximize the discriminatory power between NET G3 and NEC. Following this approach, we were able to clearly distinguish NET G3 and NEC in our cohort. On this basis, the DNA methylation results are applicable to these subgroups and provide valuable additional information not previously described in these NEN subgroups. Additional performed immunohistochemical analyses and subtyping of NET G3 and NEC cases were performed by diagnostic pathologists with a special expertise in NEN (BG and AP). We have now added the results of the additional immunohistochemical analyses in a new supplemental table, which is now included in the Supplemental Material (Table S2). Moreover, all DNA methylation patterns were reanalyzed using the new subgrouping of NET G3 and NEC.

11. Results, "NEN of different organ sites, hepatic NEN and NEN metastases exhibit distinct genetic profiles" (page 8) This subsection interrupts the methylation analysis while adding little incremental information. In lack of a better integration with the rest of the data, I recommend removing this analysis or moving most of it to supplementary information, also changing its position within the manuscript to avoid a sudden change of the focus from methylation to copy number analysis.

We have merged this section with the section before (Results, page 8). As suggested, we have re-organized this paragraph including the respective figures. In detail, we have created a new Figure 4A-F and moved the parts of the original figure to the supplemental material (Supplemental Figure S7A-D).

12. Results, "Most hepatic NEN are predicted to be of non-hepatic origin and show a foregut methylation pattern" (page 8) Again, the authors quickly dismiss tumor purity (that is, neoplastic cellularity). I recommend including it in the analysis (see point 9).

In the revised manuscript, we included our approach to select tumor areas with high tumor purity, see point 9 above and "Genomic DNA isolation" in the Materials and Methods. For DNA methylation tumor areas with high tumor cell content and low stromal and immune cell infiltration were selected. Our prediction algorithm is based on deconvolution analysis to decompose methylation data into latent methylation components (LMC). LMC were used to train Random Forest machine learning algorithms for classification of NEN subgroups. Correlation analysis of the LMCs and LUMP revealed that LMC2 correlated with LUMP and therefore, LMC2 represents immune cell infiltration (Figure 5A). Thus, we excluded methylation sites correlating with LMC2 to build our prediction algorithm, see Materials and Methods section "DNA methylation analysis using Infinium MethylationEPIC array data processing" (page 17 and 18).

13. Figure 5D: it would be nice to include also a modified version of this figure where hepatic NEN samples are placed near to the primary NEN they were predicted to originate from.

To present this important information as clearly as possible, we have improved the visibility of single data points in Figure 5. We have indicated the reference group of pancreatic NEN and the hepatic NEN predicted to originate from pancreas by color coding. Moreover, we have included an explanatory sentence in the Results section (page 10).

14. Table 4: Please add a legend to the abbreviations. The authors should explain in the results how the prediction is made when the prediction scores are similar for different sites (e.g. patient 14). Is there a cut-off or any other tool to judge the confidence of single predictions?

As suggested, we have added a legend to the abbreviations of Table 4. In addition, we included an explanatory section in the Methods (highest value determines the predicted organ site) on how the prediction is made, see page 19. In the discussion of the revised manuscript, we added the limitation that for each of the samples the highest prediction score is selected regardless of the relative height of the score compared to the other scores. This was especially the case for colorectal and gastric NEN. For example, patient #20 had the highest prediction value for gastric/duodenal NEN and second highest prediction value for colorectal NEN (page 10 and Table 4). As we further pointed out in the discussion, the relatively low number of gastric, duodenal and colonic NEN is a limitation and therefore,

follow-up studies should address this issue by analyzing larger groups of these subtypes, improving the ability to predict the primary site of NENs using DNA methylation patterns (page 14).

To further confirm our approach of using LMC-based Random Forest, we applied the XGBoost algorithm. Random Forest and XGBoost resulted in highly comparable prediction results of the NEN reference groups but only 17 of 22 (77.3%) NEN liver metastases were classified correctly (Supplemental Figure S11). These results were added on page 10 and to the methods section on page 19.

References:

1. Patel A, *et al.* Prospective, multicenter validation of a platform for rapid molecular profiling of central nervous system tumors. *Nature medicine* **31**, 1567-1577 (2025).
2. Hovestadt V ZM. Conumee: enhanced copy-number variation analysis using Illumina methylation arrays., <http://bioconductor.org/packages/conumee/> (2015).
3. Goepfert B, *et al.* Integrative analysis reveals early and distinct genetic and epigenetic changes in intraductal papillary and tubulopapillary cholangiocarcinogenesis. *Gut* **71**, 391-401 (2022).
4. Goepfert B, *et al.* Integrative Analysis Defines Distinct Prognostic Subgroups of Intrahepatic Cholangiocarcinoma. *Hepatology* **69**, 2091-2106 (2019).
5. Northcott PA, *et al.* The whole-genome landscape of medulloblastoma subtypes. *Nature* **547**, 311-317 (2017).
6. Sturm D, *et al.* New Brain Tumor Entities Emerge from Molecular Classification of CNS-PNETs. *Cell* **164**, 1060-1072 (2016).
7. Daenekas B, *et al.* Conumee 2.0: enhanced copy-number variation analysis from DNA methylation arrays for humans and mice. *Bioinformatics* **40**, (2024).
8. Taylor AM, *et al.* Genomic and Functional Approaches to Understanding Cancer Aneuploidy. *Cancer cell* **33**, 676-689 e673 (2018).
9. Martinez-Jimenez F, *et al.* Pan-cancer whole-genome comparison of primary and metastatic solid tumours. *Nature* **618**, 333-341 (2023).
10. Wang Z, *et al.* Evolving copy number gains promote tumor expansion and bolster mutational diversification. *Nature communications* **15**, 2025 (2024).
11. Watkins TBK, *et al.* Pervasive chromosomal instability and karyotype order in tumour evolution. *Nature* **587**, 126-132 (2020).
12. de Mestier L, *et al.* Molecular deciphering of primary liver neuroendocrine neoplasms confirms their distinct existence with foregut-like profile. *J Pathol* **258**, 58-68 (2022).
13. Terashima Y, *et al.* Predictive Impact of Diffuse Positivity for TTF-1 Expression in Patients Treated With Platinum-Doublet Chemotherapy Plus Immune Checkpoint Inhibitors for Advanced Nonsquamous NSCLC. *JTO Clin Res Rep* **4**, 100578 (2023).

14. Vidarsdottir H, *et al.* Comparison of Three Different TTF-1 Clones in Resected Primary Lung Cancer and Epithelial Pulmonary Metastases. *Am J Clin Pathol* **150**, 533-544 (2018).
15. Yu J, *et al.* Correlation and Comparison of Somatostatin Receptor Type 2 Immunohistochemical Scoring Systems with 68Ga-DOTATATE Positron Emission Tomography/Computed Tomography Imaging in Gastroenteropancreatic Neuroendocrine Neoplasms. *Neuroendocrinology* **112**, 358-369 (2022).
16. Muller F, *et al.* RnBeads 2.0: comprehensive analysis of DNA methylation data. *Genome Biol* **20**, 55 (2019).